nature
ecology & evolution
## OPEN

# Ecosystem productivity affected the spatiotemporal disappearance of Neanderthals in Iberia

M. Vidal-Cordasco [1]✉, D. Ocio [2], T. Hickler[3,4] and A. B. Marín-Arroyo [1]✉

What role did fluctuations play in biomass availability for secondary consumers in the disappearance of Neanderthals and the survival of modern humans? To answer this, we quantify the effects of stadial and interstadial conditions on ecosystem productivity and human spatiotemporal distribution patterns during the Middle to Upper Palaeolithic transition (50,000–30,000 calibrated years before the present) in Iberia. First, we used summed probability distribution, optimal linear estimation and Bayesian age modelling to reconstruct an updated timescale for the transition. Next, we executed a generalized dynamic vegetation model to estimate the net primary productivity. Finally, we developed a macroecological model validated with present-day observations to calculate herbivore abundance. The results indicate that, in the Eurosiberian region, the disappearance of Neanderthal groups was contemporaneous with a significant decrease in the available biomass for secondary consumers, and the arrival of the first *Homo sapiens* populations coincided with an increase in herbivore carrying capacity. During stadials, the Mediterranean region had the most stable conditions and the highest biomass of medium and medium–large herbivores. These outcomes support an ecological cause for the hiatus between the Mousterian and Aurignacian technocomplexes in Northern Iberia and the longer persistence of Neanderthals in southern latitudes.

The replacement of *Homo neanderthalensis* populations by anatomical modern humans (AMHs) is a turning point in human evolution and one of the most debated issues in Palaeolithic research. Despite the difficulties of disentangling the driving forces in such population turnover, an increasing amount of research points out that the abrupt climatic shifts during Marine Isotope Stage 3 (MIS 3; ~60–30 kyr BP) played a key role in the Middle to Upper Palaeolithic transition (MUPT) in Europe[1–4]. However, the underlying mechanisms linking climatic changes and the spatiotemporal patterns of Neanderthals' disappearance are still unknown.

The Iberian Peninsula is a key region in the exploration of the timing and causes of the Neanderthals' disappearance. Iberia is divided into two large biogeographical units: the Eurosiberian region in the north and west and the Mediterranean region in the south and east (see 'Geographic settings' in the Methods). In the Eurosiberian region, the Mousterian disappeared a few millennia earlier than in the Mediterranean region[5–7] and overlapped neither with the Châtelperronian nor the Aurignacian technocomplexes[7]. Stable isotopes[8], pollen[9] and micromammal[10] analyses suggest that towards the end of MIS 3 there was a cooling trend that led to more open landscapes, with some aridity episodes. Some authors have associated these environmental changes with the Neanderthals' decline and the resultant demographic vacuum before the early arrival of AMHs[7,10,11]. Likewise, in central areas of Iberia, the disappearance of the Mousterian and the delayed colonization by AMHs have been connected with climatic deterioration and worsening environmental conditions[1,12]. Conversely, the longer Mousterian persistence in southern latitudes has been linked to more stable climatic conditions[13–15], although their ecosystems productivity was lower than in the Eurosiberian region[16–18].

One of the biotic factors most affected by climate changes is net primary productivity (NPP). NPP is the biomass of all plant species, representing the base of the food chain for the world terrestrial ecosystems[19]. Fluctuations in NPP generate bottom-up effects that propagate through trophic levels, affecting the abundance of both primary and secondary consumers[20]. Different studies have demonstrated the significance of NPP for hunter-gatherer demographics[21,22] and the relevance of the herbivore biomass for human evolution throughout the Pleistocene[23–25]. Consequently, ecosystems' carrying capacity should be considered in the debate of the demise of Neanderthal populations[26]. Nevertheless, the impact of the stadial–interstadial cycles on the plant and herbivore biomass, as well as the influence of the ecosystems' productivity on the spatial and temporal replacement patterns of Neanderthals by AMH, remain primarily unexplored hitherto.

The aims of the current study were: (1) to quantify the effects of the MIS 3 stadial and interstadial conditions on the plant and herbivore biomass; and (2) to test whether the temporal and spatial patterns of the MUPT in Iberia were affected by alterations in ecosystem productivity. To this end, we integrated three modelling approaches. First, we used optimal linear estimation (OLE) models, summed probability distribution (SPD) of archaeological assemblages and Bayesian age modelling to reconstruct an updated timescale for the MUPT in the four biogeographical regions of Iberia. Second, we used a generalized dynamic vegetation model with tested climate inputs from an atmospheric general circulation model to estimate the evolution of NPP between 55 and 30 kyr BP in each archaeological and palaeontological site of the MIS 3. Finally, we validated a macroecological model against empirical present-day herbivore densities from a broad range of terrestrial ecosystems and used this modelling approach to estimate the herbivore abundance in each MIS 3 stadial and interstadial phase.

## Results

**Biogeographical differences in herbivore and plant biomass.** The analyses performed in the current study focused on the four

[1]Grupo I+D+i EvoAdapta (Evolución Humana y Adaptaciones Económicas y Ecológicas durante la Prehistoria), Departamento Ciencias Históricas, Universidad de Cantabria, Santander, Spain. [2]Mott MacDonald, Cambridge, UK. [3]Senckenberg Biodiversity and Climate Research Centre (SBiK-F), Frankfurt am Main, Germany. [4]Department of Physical Geography, Goethe University, Frankfurt, Germany. ✉e-mail: vidalma@unican.es; marinab@unican.es

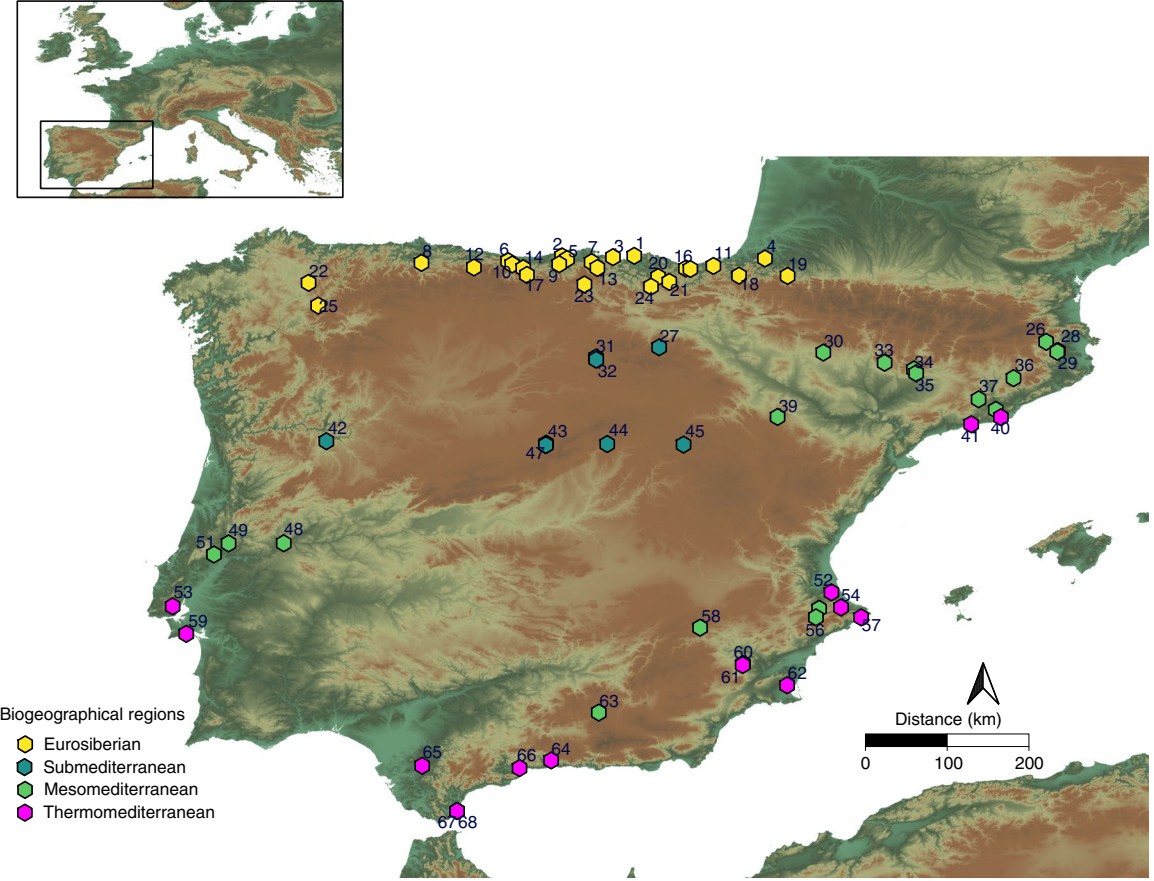

**Fig. 1 | Locations of the archaeological and palaeontological sites where NPP was estimated.** The colours show the classification of each site according to the present-day biogeographical classification. 1, Aranbaltza; 2, Covalejos; 3, El Cuco; 4, Isturitz; 5, Cueva Morín; 6, Llonín; 7, Cobrante; 8, La Viña; 9, El Castillo; 10, Esquilleu; 11, Aitzbitarte III; 12, La Güelga; 13, El Mirón; 14, Jou Puerta; 15, Ekain; 16, Amalda; 17, Cowshead; 18, Mainea; 19, Gatzarria; 20, Bolinkoba; 21, Labeko Koba; 22, Cova da Valiña; 23, Prado Vargas; 24, Arrillor; 25, Cova Eirós; 26, Els Ermitons; 27, Peña Miel; 28, Mollet I; 29, Arbreda; 30, Fuentes de San Cristóbal; 31, Cueva Millán; 32, La Ermita; 33, Los Moros I; 34, Cova Gran; 35, Roca dels Bous; 36, Teixoneres; 37, Abric Romaní; 38, Coll Verdaguer; 39, Aguilón P5; 40, Riera dels Canyars; 41, Cova Foradada; 42, Salto do Boi; 43, Abrigo del Molino; 44, Jarama; 45, Casares; 46, Cueva de la Zarzamora; 47, Portalón del Tejadilla; 48, Foz do Enxarrique; 49, Gruta do Caldeirao; 50, Lapa do Picareiro; 51, Gruta da Oliveira; 52, Cova de les Malladetes; 53, Pego do Diabo; 54, Cova Foradada; 55, Cova Beneito; 56, El Salt; 57, Cova de les Cenderes; 58, El Niño; 59, Figueira Brava; 60, La Boja; 61, Cueva Antón; 62, Sima de las Palomas; 63, Carihuela; 64, Zafarraya Cueva del Boquete; 65, Cueva del Higueral de Valleja; 66, Bajondillo; 67, Gorham's Cave; 68, Vanguard Cave.

main biogeographical regions of Iberia: the Eurosiberian region in the temperate areas of the North; the Supramediterranean in the Northern Plateau and Iberian System; and the two bio-climatic belts that make up the Mediterranean region (the Mesomediterranean and the Thermomediterranean regions (for details, see the 'Geographic settings' subsection in the Methods)) (Fig. 1). In the interstadial periods, the highest NPP mean is observed in the Eurosiberian region (0.33 kg km$^{-2}$ yr$^{-1}$), followed by the Mesomediterranean (0.31 kg km$^{-2}$ yr$^{-1}$), Supramediterranean (0.26 kg km$^{-2}$ yr$^{-1}$) and Thermomediterranean (0.25 kg km$^{-2}$ yr$^{-1}$) regions (Table 1). During the stadial periods, the only bio-geographical region where NPP did not decrease significantly was the Thermomediterranean region (Table 1). In the cold stadial phases, NPP was highest in the Mesomediterranean (0.30 kg km$^{-2}$ yr$^{-1}$), followed by the Eurosiberian (0.27 kg km$^{-2}$ yr$^{-1}$), Thermomediterranean (0.25 kg km$^{-2}$ yr$^{-1}$) and Supramediterranean (0.20 kg km$^{-2}$ yr$^{-1}$) regions (Table 1). The sharpest fluctuations of NPP are observed in the Eurosiberian and Supramediterranean regions, where NPP decreased, on average, 0.06 kg km$^{-2}$ yr$^{-1}$ in sta-dial times and where the coefficient of variation of NPP was twice as high as during the interstadial phases (Table 1). Conversely, in the Mesomediterranean and Thermomediterranean regions, NPP

decreased by about 0.01 kg km$^{-2}$ yr$^{-1}$ on average during the stadial periods (Table 1).

Three main clusters were identified in the evolution of NPP in the Iberian Peninsula (Fig. 2). NPP estimated in 64% of the archaeo-palaeontological sites located in the Eurosiberian region was grouped into the first cluster, 58% of NPP estimated in the sites located in the Mesomediterranean was grouped into the sec-ond cluster and 74% of NPP estimated in the sites located in the Thermomediterranean region was grouped into the third cluster (Supplementary Table 1). However, the temporal evolution of NPP estimated in the sites located in ecotones did not differ from these three main groups (Supplementary Fig. 1). Accordingly, the only biogeographical region in which NPP did not evolve differently during MIS 3 was the Supramediterranean region (Supplementary Fig. 1). These results indicate that the Eurosiberian, Mesomediter-ranean and Thermomediterranean regions experienced different trends in NPP during MIS 3.

According to the Jaccard similarity index (JSI), the herbivore guild composition differed between the Eurosiberian, Supramediter-ranean, Mesomediterranean and Thermomediterranean regions (Supplementary Fig. 2). The biomass of large herbivores was sub-stantially higher in the Eurosiberian region in both the stadial

**Table 1 | NPP in each biogeographical region during the stadial and interstadial periods of the late MIS 3**

| Region | Period | Mean (kg km⁻² yr⁻¹) | s.d. | Coefficient of variation | Wilcoxon rank-sum test | |
|---|---|---|---|---|---|---|
| | | | | | W | P |
| Eurosiberian | Stadial | 0.271 | 0.070 | 25.92 | 7,507 | $9.9 \times 10^{-14}$ |
| | Interstadial | 0.339 | 0.039 | 11.33 | | |
| Supramediterranean | Stadial | 0.201 | 0.044 | 22.26 | 8,244 | $2.2 \times 10^{-16}$ |
| | Interstadial | 0.268 | 0.028 | 10.77 | | |
| Mesomediterranean | Stadial | 0.300 | 0.030 | 9.99 | 5,799 | 0.002 |
| | Interstadial | 0.312 | 0.026 | 8.58 | | |
| Thermomediterranean | Stadial | 0.259 | 0.023 | 9.09 | 4,621 | 0.986 |
| | Interstadial | 0.258 | 0.027 | 10.65 | | |

The Wilcoxon rank-sum test evaluated whether the estimated NPP in each biogeographical region was significantly different ($P < 0.05$) during the stadial and interstadial times.

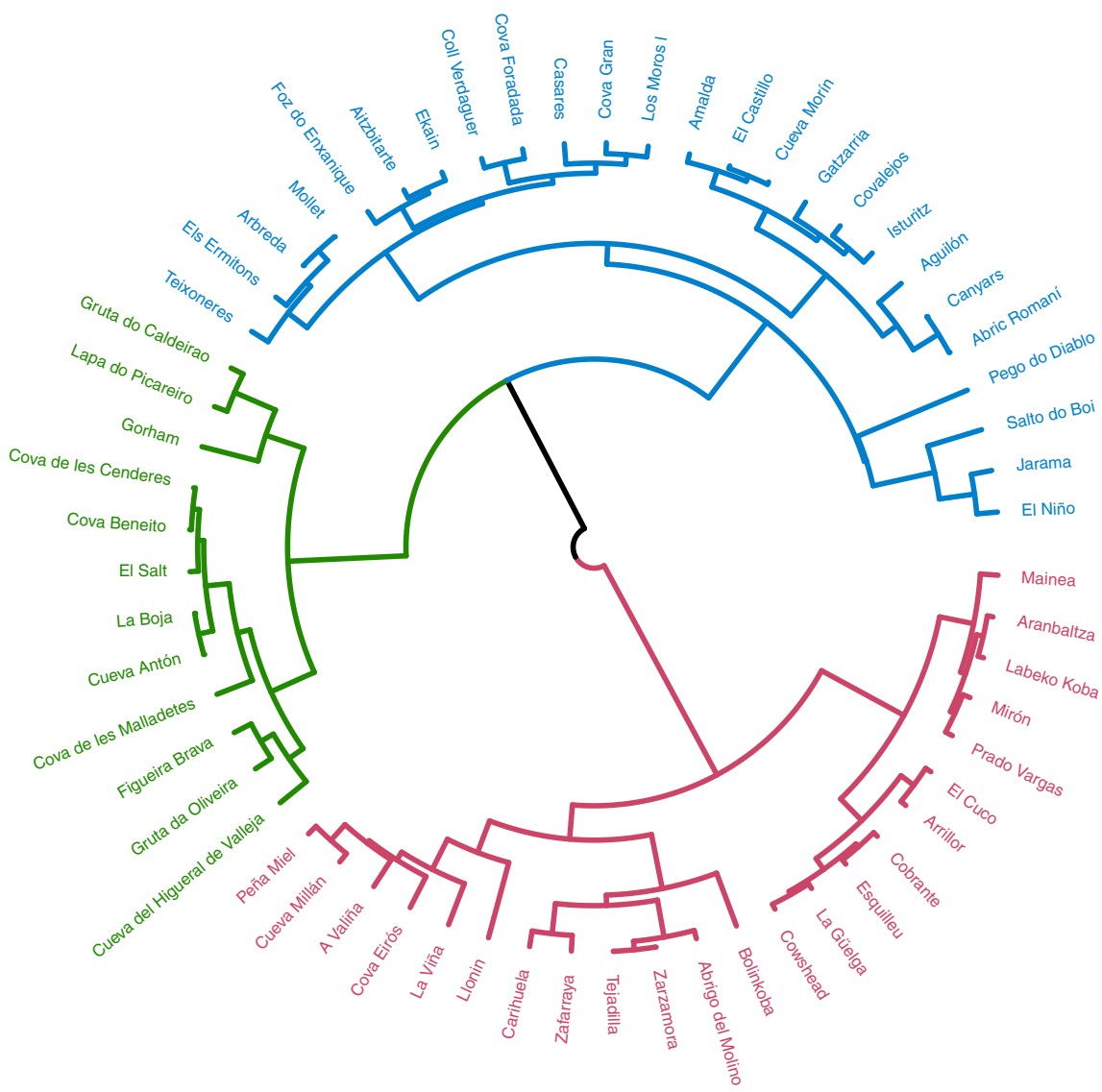

**Fig. 2 | Circular dendrogram showing the clustering association of NPP estimated in the surrounding area of each archaeo-palaeontological site based on the *d*CORT dissimilarity index.** Each colour indicates one cluster. Red, first cluster (NPP estimated in 64% of the sites located in the Eurosiberian region). Blue, second cluster (NPP estimated in 58% of the sites included in the Mesomediterranean region). Green, third cluster (NPP estimated in 74% of the sites located in the Thermomediterranean region).

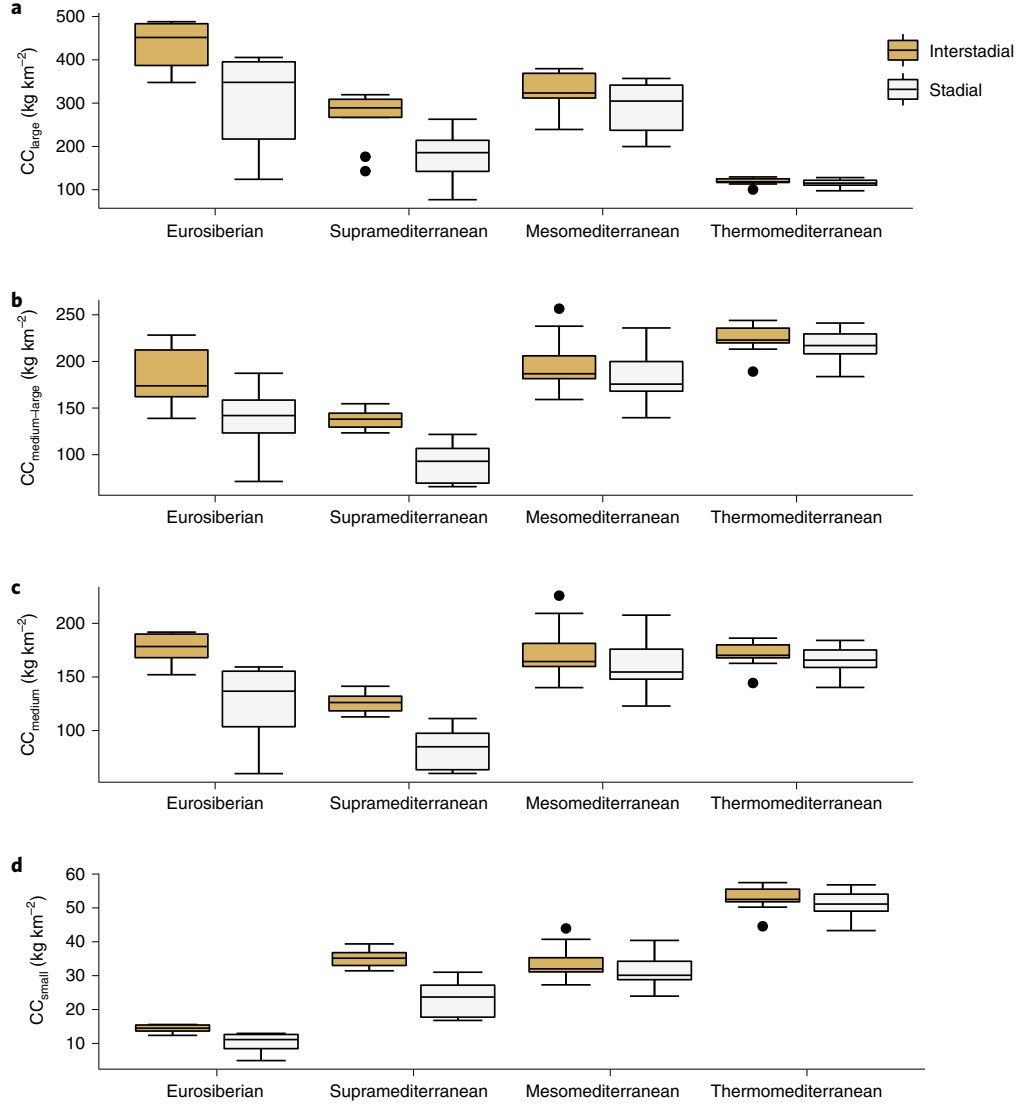

**Fig. 3 | Carrying capacity of different sized herbivore species during the stadial and interstadial periods. a–d,** Box and whisker plots showing the carrying capacity of large (>500 kg; **a**), medium–large (100–500 kg; **b**), medium (10–100 kg; **c**) and small (<10 kg; **d**) herbivore species during the stadial and interstadial periods. The centre line of each plot corresponds to the median, the box edges correspond to the first and third quartiles, the whiskers correspond to 1.5× the interquartile range and the dots represent outliers. The estimated carrying capacity in each specific stadial and interstadial phase, along with the 95% CI of the estimation, is provided in Supplementary Table 2.

and interstadial periods, and the lowest biomass of the large herbivores was observed in the Thermomediterranean region (Fig. 3). In contrast, the highest biomass of small herbivores was always observed in the Thermomediterranean region and the lowest values were systematically observed in the Eurosiberian region (Fig. 3) (Supplementary Table 2). Thus, a latitudinal gradient was identified in the biomass distribution of large and small herbivores in the Iberian Peninsula during MIS 3 (Fig. 3). However, among these biogeographical regions, the differences in the biomass of medium-sized herbivores changed during the stadial and interstadial phases. In the interstadial periods, the highest biomass of medium-sized herbivores was observed in the Eurosiberian region (177.27 kg km⁻² yr⁻¹), followed by the Mesomediterranean (175.21 kg km⁻² yr⁻¹), Thermomediterranean (170.93 kg km⁻² yr⁻¹) and Supramediterranean regions (125.85 kg km⁻² yr⁻¹) (Fig. 3). In the stadial phases, the biomass of medium-sized herbivores decreased by up to one-third in the Eurosiberian (126.5 kg km⁻² yr⁻¹) and Supramediterranean (84.3 kg km⁻² yr⁻¹) regions, whereas in

the Mesomediterranean it decreased by 8% (161.74 kg km⁻² yr⁻¹) and in the Thermomediterranean it decreased, on average, by 3% (165.57 kg km⁻² yr⁻¹) (Fig. 3). Consequently, during the cold stadial periods, the biomass of medium-sized herbivores was meaningfully higher in the Mediterranean region (Fig. 3).

**Ecosystem productivity and disappearance of Neanderthals.** The results obtained from the Bayesian and OLE models are summarized in Table 2. The temporal evolution of NPP and SPD of archaeological assemblages are summarized in Fig. 4. The herbivore biomass in each stadial and interstadial phase of the MIS 3 is shown in Fig. 5. The frequency of Mousterian assemblages decreased around 45 kyr ʙᴘ and disappeared during Greenland Stadial 12 (GS-12) according to the SPD (Fig. 4), which is consistent with the extinction date estimated by the OLE model and the probability distribution function at a 95% confidence interval (CI) obtained from the Bayesian age models (BAMs; Table 2). The onset of GS-12 in the Eurosiberian region corresponds to a decrease of 33.3% in

**Table 2 | Median, lower and upper bounds of each model's 95% CI for the start and end of the Mousterian, Châtelperronian and Aurignacian technocomplexes in each biogeographical region according to the BAMs and OLE model**

| Region | Culture | Boundary | BAMs | | | OLE | | | | | |
|---|---|---|---|---|---|---|---|---|---|---|---|
| | | | | | | Central range | | | Resampling | | |
| | | | Median | Lower bound | Upper bound | Median | Lower bound | Upper bound | Median | Lower bound | Upper bound |
| Eurosiberian | Mousterian | End | 45.04 | 44.06 | 47.08 | 45.03 | 43.88 | 45.29 | 44.56 | 43.06 | 44.96 |
| | Châtelperronian[a] | Start | 42.31 | 42.02 | 43.83 | 44.13 | 43.62 | 53.20 | 48.27 | 47.38 | 65.87 |
| | | End | 41.94 | 41.02 | 42.34 | 41.75 | 41.04 | 41.88 | 40.61 | 37.65 | 41.03 |
| | Aurignacian | Start | 42.56 | 42.05 | 43.33 | 44.47 | 43.97 | 46.64 | 45.93 | 45.25 | 51.22 |
| | | End | 35.43 | 34.73 | 36.01 | 35.29 | 35.02 | 35.36 | 35.07 | 34.61 | 35.20 |
| Supramediterranean | Mousterian[a] | End | 41.95 | 40.48 | 42.76 | 41.63 | 39.61 | 41.93 | 41.31 | 38.19 | 41.79 |
| Mesomediterranean | Mousterian | End | 41.31 | 40.80 | 41.88 | 40.86 | 39.87 | 41.16 | 40.86 | 39.88 | 41.14 |
| | Aurignacian | Start | 41.98 | 41.48 | 42.62 | 42.77 | 42.40 | 44.10 | 42.93 | 42.55 | 44.47 |
| | | End | 36.36 | 35.69 | 36.97 | 36.12 | 35.54 | 36.25 | 35.94 | 35.24 | 36.13 |
| Thermomediterranean | Mousterian | End | 35.68 | 32.64 | 36.62 | 34.95 | 32.87 | 35.43 | 34.96 | 32.92 | 35.44 |
| | Châtelperronian[a] | Start | 40.07 | 39.28 | 42.76 | – | – | – | – | – | – |
| | | End | 39.29 | 36.37 | 40.24 | – | – | – | – | – | – |
| | Aurignacian | Start | 41.89 | 41.44 | 42.45 | 41.02 | 40.31 | 43.52 | 41.40 | 40.58 | 44.52 |
| | | End | 33.44 | 32.74 | 33.95 | 30.27 | 28.92 | 30.70 | 30.28 | 28.95 | 30.69 |

[a]The sample size was lower than 10. For details, see Supplementary Note 1 in the Supplementary Information. The OLE approach was not applied to the Châtelperronian in the Thermomediterranean region because it was represented by only one assemblage. For details, see 'Chronological assessment' in the Methods.

NPP (Fig. 4) and a decrease of 45% in herbivore biomass (Fig. 5). Despite the higher uncertainty in estimating the first appearance of the Châtelperronian and Aurignacian in this region, the number of Châtelperronian and Aurignacian assemblages substantially increased with the rapid recovery of ecosystem productivity during Greenland Interstadial 11 (GI-11) (Fig. 4). Between GI-11 and GI-7, the fluctuations in NPP and herbivore carrying capacity were smoother in the stadial–interstadial transitions. Nonetheless, after GI-7, NPP started another cycle of important fluctuations, which coincided with the end of the Aurignacian in this region.

In the Supramediterranean region, the Mousterian disappeared between 41.93 and 39.61 kyr cal BP according to the OLE method, consistent with the chronological range obtained from the BAMs. According to the SPD, the frequency of Mousterian assemblages decreased during GS-11 and disappeared during GS-10. During GS-11, NPP decreased by 21.42% (Fig. 4) and the herbivore carrying capacity decreased by 25.2% (Fig. 5). In GS-10, NPP decreased, on average, by 27.11% in the Supramediterranean region (Fig. 4) and the herbivore carrying capacity decreased by 35.69% (Fig. 5). After that, neither Châtelperronian nor Aurignacian occupations are documented in this region during the remainder of the late MIS 3.

According to the OLE method, in the Mesomediterranean region, the Mousterian technocomplex disappeared between 41.16 and 39.87 kyr cal BP and the first arrival of the Aurignacian technocomplex took place between 44.1 and 42.4 kyr cal BP. These outcomes are consistent with those obtained from the BAMs (Table 2). Unlike the Eurosiberian and Supramediterranean regions, in the Mesomediterranean region there was a chronological overlap of at least 400 years between the Mousterian and Aurignacian. During the MUPT, NPP fluctuated, on average, by 7.8% between the stadial and interstadial periods, and the herbivore carrying capacity fluctuated by 8.9% and thus remained considerably stable. In the Thermomediterranean region, the Mousterian culture disappeared between 35.43 and 32.87 kyr cal BP and the start of the Aurignacian was between 43.52 and 40.31 kyr cal BP (Table 2). Accordingly, the Thermomediterranean was the region with the longest potential overlap between *H. neanderthalensis* and *H. sapiens*, with the most constant conditions in NPP and the highest biomass of medium and medium–large herbivores (Figs. 4 and 5). Neither in the Mesomediterranean nor in the Thermomediterranean regions could the disappearance of the Mousterian be associated with a remarkable fluctuation in ecosystem productivity (Figs. 4 and 5).

## Discussion
The relationships between climatic changes and species demographic trends include many different potential links, which commonly hampers the identification of specific causal mechanisms in the disappearance of extinct species. The current study provides robust evidence about the impact of MIS 3 stadial–interstadial cycles on ecosystem productivity and its coincidence with the spatial and temporal replacement patterns of Neanderthals by AMH in Iberia.

The results obtained show that the end of the Mousterian was coeval with a significant decrease in NPP and herbivore carrying capacity in the Eurosiberian and Supramediterranean regions. In turn, Neanderthals survived longer in the Mesomediterranean and Thermomediterranean regions where NPP fluctuations were small. These results reinforce the hypothesis that the previously proposed demographic hiatus[7,27] between the late Mousterian and the Châtelperronian and Aurignacian technocomplexes in the Eurosiberian region was motivated by a significant decrease in the trophic resource availability for secondary consumers, which affected late Neanderthal populations. In addition, the arrival of the Châtelperronian and Aurignacian technocomplexes coincided with rapid recovery of the herbivore biomass in GI-11, and the disappearance of the Aurignacian occurred with another remarkable decrease in ecosystem productivity, which would suggest that fluctuations in NPP and herbivore biomass affected Neanderthals and AMHs similarly. This observation remains independent of the authorship of the Châtelperronian. It has been proposed that the appearance of the Châtelperronian technocomplex in the Cantabrian Region may indicate that Neanderthal populations arrived from southwestern

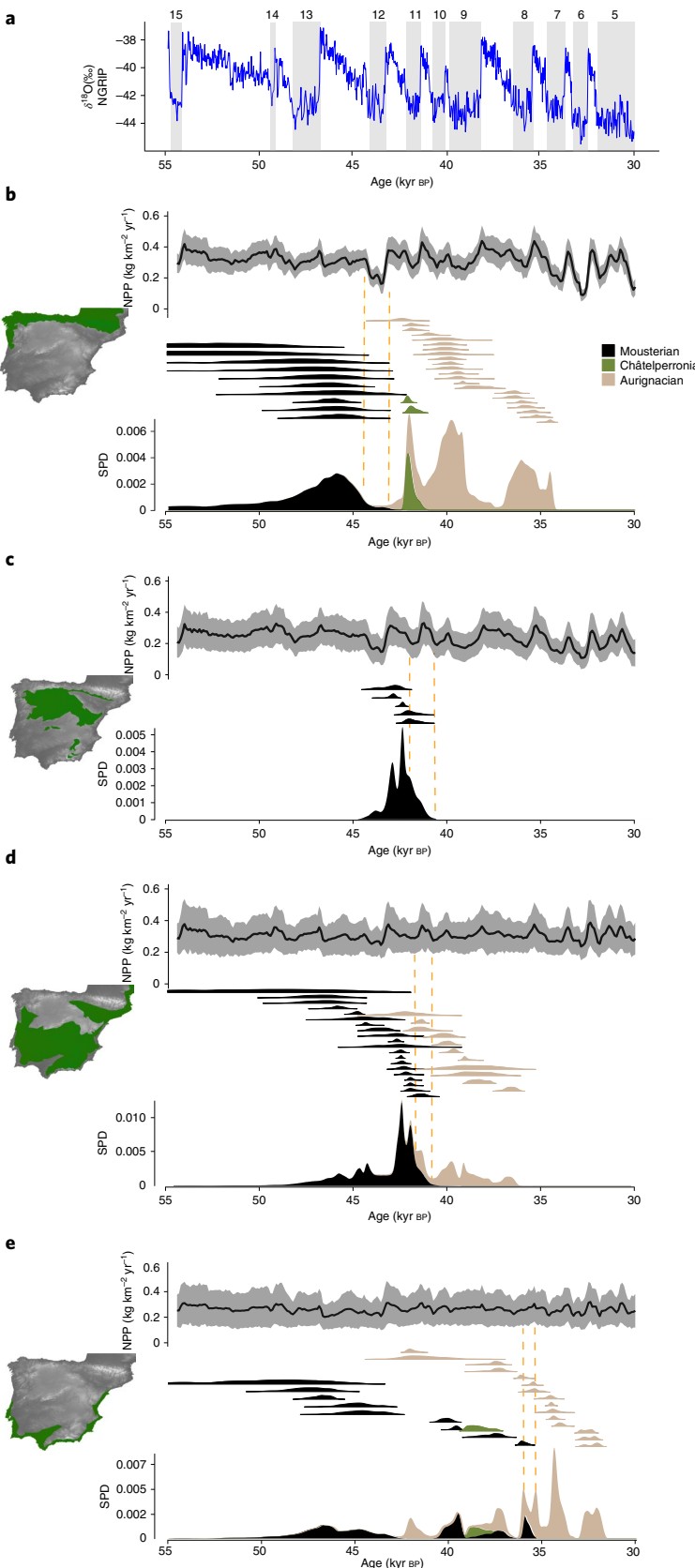

**Fig. 4 | δ¹⁸O, NPP and chronology of the Middle to Upper Paleolithic transition. a**, $\delta^{18}$O between 54.5 and 30 kyr BP, as measured by the North Greenland Ice Core Project (NGRIP). Vertical grey numbered bars represent the stadial phases. **b–d**, Top, temporal evolution of NPP in the Eurosiberian (**b**), Supramediterranean (**c**), Mesomediterranean (**d**) and Thermomediterranean (**e**) regions over the same timescale as shown in **a**. Black lines show mean values while shaded grey areas represent 2 × s.d. Middle, age probability distributions of the archaeological assemblages in these regions at 95% CI. Bottom, SPD of these assemblages. The vertical yellow dashed lines show the time range of the end of the Mousterian in each region.

France due to pressure from expanding populations of AMHs[7]; however, some authors cast doubts on the Châtelperronian–Neanderthal association[28]. Despite this, the results obtained show that both species were similarly affected by important fluctuations in herbivore and plant biomass. A recent zooarchaeological assessment has shown that Neanderthals and AMHs exploited the same biotic resources during the MUPT in the Cantabrian Region[29], consistent with the results of our analyses.

The Ebro frontier hypothesis claims that once Neanderthal populations were relegated to southern latitudes of Iberia due to climate worsening there was a natural barrier with semi-desert conditions between the Eurosiberian and Mediterranean biogeographical regions, which segregated southern Neanderthal populations from northern AMHs[30,31]. This study shows that the NPP of the archaeo-palaeontological sites located in the ecotones between the Eurosiberian and Mediterranean regions did not evolve differently from the NPP estimated in the Eurosiberian or Mediterranean regions. However, NPP in the inner areas of central Iberia (that is, the transitional Supramediterranean region) was significantly lower, with specific moments of substantial reductions in ecosystem productivity (Table 1). Recent studies provide evidence for an early arrival of AMHs in the Mediterranean[32] and Atlantic[33] rims of Iberia, thus contradicting the Ebro frontier hypothesis. Besides this, the results obtained in this study indicate that if such a barrier existed it would have been in the inner areas of central Iberia rather than the north–south division from the Ebro basin. Moreover, the Mediterranean and Atlantic coastal margins held high NPP throughout MIS 3, likely with high habitat suitability for modern human populations. This fact could have facilitated their rapid dispersal across Iberia.

Most of the late Mousterian sites located in the hinterland territories of Iberia have been associated with short-term occupations abandoned during climate worsening events[12,34]. Many of these sites are close to the natural corridors of the Mesomediterranean biogeographical region (for example, the Ebro valley or Duero valley). Thus, the persistence of Neanderthal populations in the Supramediterranean region might be related to movements to inner areas during mild conditions, as suggested by previous research[34,35]. This idea is reinforced by the fact that the disappearance of the Mousterian in the Supramediterranean region coincided with its disappearance in the Mesomediterranean region. In addition, the meaningfully low NPP and remarkable fluctuations in herbivore carrying capacity in the inner areas of Iberia could be essential for our understanding of not only the low and scattered population of the hinterland during the MUPT[36] but also the late arrival of AMHs to this region[37].

It has been proposed that grasslands of the Eurosiberian region had higher primary productivity and, consequently, could sustain more large herbivores (for example, mammoths or woolly rhinoceroses) than Mediterranean areas[16,18,38]. Moreover, an increasing number of studies have shown that small prey, with high capture costs and low energetic return rates, were exploited throughout the Middle and Upper Palaeolithic in the Mediterranean region of Iberia[39,40]. Therefore, some authors have proposed that Neanderthal populations were forced towards environments with low biomass availability[16,18]. The results obtained in this study confirm that NPP was higher in the Eurosiberian region than in the Mediterranean region, which probably allowed the higher diversity of large herbivore species observed in the northern latitudes. However, in the Mediterranean areas, the carrying capacity of medium and medium–large herbivores was similar to that of the Eurosiberian region under interstadial conditions, and higher during stadial times. This is explained by the combination of two specific factors: the significant decrease of NPP in the Eurosiberian region and the lower number of large species in the Mediterranean region. Different studies have shown that large herbivores in extant

ecosystems are associated with lower densities of smaller herbivore species because they use a substantial proportion of NPP[41–43]. Therefore, if not only NPP but also the herbivore guild structure is considered, the carrying capacity of medium and medium–large herbivores will be higher in Mediterranean regions during the cold stadial phases of MIS 3. Previous studies have shown that medium and medium–large herbivores made up the bulk of Neanderthal and AMH diets[29], so fluctuations in their prey abundance could have impacted their survival costs. Hence, the results would not support the contention that the last Neanderthal populations retreated to impoverished environments. In contrast, the persistence of Neanderthal populations in southern latitudes might have been motivated by the higher biomass of medium and medium–large herbivores of the Thermomediterranean ecosystems during stadial conditions, which would support that southern Iberia acted as a refugium for Neanderthal populations during the cold stages of the late MIS 3 (refs. [2,44]).

Ecosystem productivity fluctuated in both the Mediterranean and Eurosiberian regions, but in the Mediterranean areas these oscillations were more minor. This stability is supported by different palaeoenvironmental records revealing that the Mediterranean belt was less affected by MIS 3 cooling events than the Eurosiberian region[13,15,45]. Furthermore, the more stable evolution of NPP may become essential for our understanding of structural continuity in the herbivore guild composition in the Mediterranean region throughout the Late Pleistocene[46,47], as well as the megafauna extinctions in northern latitudes during the Late Quaternary. Ancient DNA studies indicate that the genetic diversity of several herbivore species decreased during 50–30 kyr BP in different Eurosiberian regions[48]. The results obtained in this study show that these herbivore population changes during the late MIS 3 could be primarily explained by fluctuations in NPP.

It should be noted that this study does not prove causality between environmental changes and Neanderthal population demise but provides evidence suggesting that there was a significant decrease in ecosystem productivity during the MUPT in the Eurosiberian region. SPD results indicate that the decrease in the Neanderthal population probably started during GI-12 and ended during GS-12. Therefore, this decrease in ecosystem productivity probably affected the last Neanderthals in this region, but it cannot be interpreted as a direct and single cause of their demise. In contrast, in the Mesomediterranean and Thermomediterranean regions, the disappearance of the Mousterian and the appearance of the Aurignacian cannot be related to changes in the availability of trophic resources for secondary consumers, which suggests that the causes of the demise of Neanderthal populations probably differed between regions within Iberia. However, the common and rapid fluctuations in ecosystem productivity in the Eurosiberian region and inner areas of Iberia could have impacted on inter-population connectivity of Neanderthal populations and generated gene pool bottlenecking events[49]. Therefore, these fluctuations in plant and herbivore biomass may be essential for our understanding of not only the retreatment of Neanderthal populations to southern latitudes but also their posterior extinction.

An alternative hypothesis for the disappearance of Neanderthals could be competition with AMHs for the same resources[50]. This study shows that this hypothesis could only be supported in the Eurosiberian region if the authorship of the Châtelperronian corresponds to *H. neanderthalensis* as has been recently proposed[27]. In contrast, in the central areas of Iberia, this hypothesis cannot be maintained with the current evidence. In turn, in the Mesomediterranean and Thermomediterranean regions, the most extended overlap between AMHs and Neanderthals and the stable evolution of ecosystem productivity in MIS 3 suggest that a competition hypothesis would be plausible. However, this hypothesis requires further research.

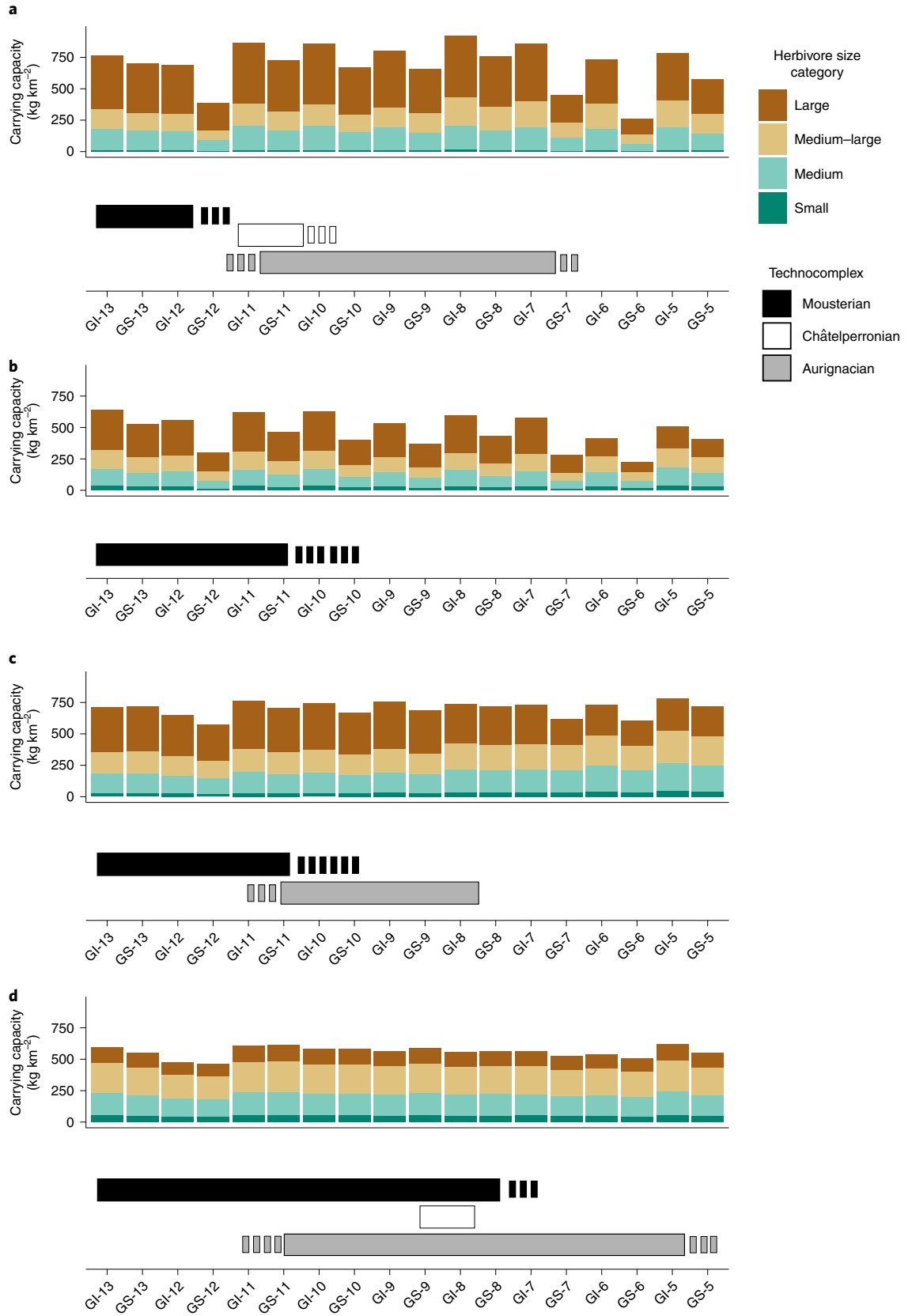

**Fig. 5 | Carrying capacities of different sized herbivore species during the MIS 3 stadial and interstadial periods. a–d**, Carrying capacity by herbivore size (top), and technocomplex duration (bottom) in the Eurosiberian (**a**), Supramediterranean (**b**), Mesomediterranean (**c**) and Thermomediterranean (**d**) regions during the MIS 3 stadial and interstadial periods. The horizontal bars show the duration of each technocomplex according to the SPD, with each dashed part showing the time interval between the start of the decrease in the frequency of archaeological assemblages and the disappearance of the culture. The herbivore size categories were as described in the Fig. 3 caption.

 

Although the results obtained for MUPT chronology are, overall, consistent between the Bayesian and OLE models, some caveats should be noted. First, the outcomes obtained here are contingent on the availability of new sites and dates or re-dating programmes. Second, each of these methods has its assumptions and limitations (see Methods for details). Despite the consistency in the outcomes obtained from them, the OLE approach shows substantial uncertainty in dating the start of some technocomplexes, particularly when the number of chronometric determinations is low and the standard errors of the calibrated dates are large and close to the limit of the calibration curve. Third, despite the match between the fluctuations in NPP and the stadial–interstadial phases, the time intervals of these stadial–interstadial periods are also subject to some chronological uncertainty[51]. Despite the popularity of SPD, it should be noted that the frequency of archaeological assemblages is not a direct indicator of population size; therefore, it should not be used as a pristine demographic proxy[52] (see Methods for details). Notwithstanding this, the conclusions obtained in this paper uphold even considering the current uncertainty in the timings of the first and last appearance of each culture. In this connection, the comparison of the raw calibrated dates (when no modelling method is used) with the reconstructed NPP in each region suggests that the conclusions of this study do not depend on the modelling approach used to estimate the chronology of the MUPT (Supplementary Fig. 3). Regarding the other modelling approaches used in this study, the current macroecological model used to estimate herbivore abundance showed good model performance, but further studies should expand on some methodological aspects, such as the spatial distribution of the estimated carrying capacity at finer geographic scales, the relevance of the abundance of specific herbivore species for Neanderthal and AMH subsistence strategies, or the demand exerted by secondary consumers. Nevertheless, the model presented here has successfully identified a mechanism that explains the spatiotemporal replacement patterns of *H. neanderthalensis* by AMHs in Iberia.

## Methods

**Fauna, culture and chronology datasets.** A geo-referenced dataset of chronometric dates covering the late MIS 3 (55–30 kyr cal BP) was compiled from the literature (dataset 1). The dataset included 363 radiocarbon, thermoluminescence, optically stimulated luminescence and uranium series dates obtained from 62 archaeological sites and seven palaeontological sites. These chronological determinations were obtained from ten palaeontological levels and 138 archaeological levels. The archaeological levels were culturally attributed to the Mousterian ($n=75$), Châtelperronian ($n=6$) and Aurignacian ($n=57$) technocomplexes. A number of issues can potentially hamper the chronological assessment of Palaeolithic technocomplexes from radiocarbon dates, such as pretreatment protocols that do not remove sufficient contaminants or the quality of the bone collagen extracted. Moreover, discrepancies in cultural attributions or stratigraphic inconsistencies are commonly detected in Palaeolithic archaeology. Information regarding the quality of date determinations and cultural attribution or stratigraphic issues is provided in the Supplementary Information.

Our dataset also included the presence of herbivore species recovered from each archaeo-palaeontological site (hereafter referred to as local faunal assemblages (LFAs)), their body masses and their chronology. The mean body mass of both sexes, for each species, was obtained from the PHYLACINE database[53] and used in the macroecological modelling approach described below (see 'Carrying capacity of herbivores'). For visual representation purposes, the herbivore species were grouped into four weight categories: small (<10 kg), medium (10–100 kg), medium–large (100–500 kg) and large (>500 kg). The chronology of the occurrence of each herbivore species was assumed to be the same as the dated archaeo-palaeontological layer where the species remains were recovered. Thus, to estimate the chronological range of each species in each region, all radiocarbon determinations were calibrated with the IntCal20 calibration curve[54] and OxCAL4.2 software[55]. The BAMs were run to compute the upper and lower chronological boundaries at a CI of 95.4% of each LFA (see 'Chronological assessment' for more details). One of the purposes of the current study was to estimate the potential fluctuations in herbivore biomass during the stadial and interstadial periods of the late MIS 3. Accordingly, the time spans of the LFAs were classified into the discrete GS and GI phases provided by Rasmussen et al.[51].

**Geographic settings.** The Iberian Peninsula locates at the southwestern edge of Europe (Fig. 1). It constitutes a large geographic area that exhibits a remarkable

diversity of ecosystems, climates and landscapes. Both now and in the past, altitudinal, latitudinal and oceanic gradients affected the conformation of two biogeographical macroregions with different flora and fauna species pools: the Eurosiberian and Mediterranean regions[13,46]. In the north, along the Pyrenees and Cantabrian strip, the Eurosiberian region is characterized by oceanic influence and mild temperatures in the present day, whereas the Mediterranean region features drier summers and milder winters (Fig. 1). Between the Eurosiberian and Mediterranean regions, there is a transitional area termed Submediterranean or Supramediterranean. Lastly, the Mediterranean region is divided into two distinctive bioclimatic belts: (1) the Thermomediterranean region, located at lower latitudes, with high evapotranspiration rates and affected by its proximity to the coast; and (2) the Mesomediterranean region, with lower temperatures and wetter conditions (Fig. 1).

Previous studies have shown that zoocoenosis and phytocoenosis differed between these macroregions in the Pleistocene[13,46]. However, flora and fauna distributions changed during the stadial–interstadial cycles in the Iberian Peninsula, which suggests potential alterations in the boundaries of these biogeographical regions. The modelling approach used in this study to estimate the biomass of primary consumers is dependent on the reconstructed NPP and the herbivore guild structure in each biogeographical region. To test the suitability of the present-day biogeographical demarcations of the Iberian Peninsula during MIS 3, we assessed whether the temporal trends of NPP and the composition of each herbivore palaeocommunity differed between these biogeographical regions during the MUPT.

Chouakria and Nagabhushan[56] proposed a dissimilarity index to compare time series data by taking into consideration the proximity of values and the temporal correlation of the time series:

$$\text{CORT}(S_1, S_2) = \frac{\sum_{i=1}^{p-1} \left(u_{(i+1)} - u_i\right)\left(v_{(i+1)} - v_i\right)}{\sqrt{\sum_{i=1}^{p-1} \left(u_{(i+1)} - u_i\right)^2}\sqrt{\sum_{i=1}^{p-1} \left(v_{(i+1)} - v\right)^2}} \quad (1)$$

where $S_1$ and $S_2$ are the time series of data, $u$ and $v$ represent the values of $S_1$ and $S_2$, respectively, and $p$ is the length of values of each time series. $\text{CORT}(S_1, S_2)$ belongs to the interval $(-1,1)$. The value $\text{CORT}(S_1, S_2)=1$ indicates that in any observed period $(t_i, t_{i+1})$, the values of the sequence $S_1$ and those of $S_2$ increase or decrease at the same rate, whereas $\text{CORT}=-1$ indicates that when $S_1$ increases, $S_2$ decreases or vice versa. Lastly, $\text{CORT}(S_1, S_2)=0$ indicates that the observed trends in $S_1$ are independent of those observed in $S_2$. To complement this approach by considering not only the temporal correlation between each pair of time series but also the proximity between the raw values, these authors proposed an adaptive tuning function defined as follows:

$$d\text{CORT}(S_1, S_2) = f(\text{CORT}(S_1, S_2)) \times d(S_1, S_2) \quad (2)$$

where

$$f(x) = \frac{2}{1 + \exp(kx)}, k \geq 0 \quad (3)$$

In this study, $k$ was 2, meaning that the behaviour contribution was 76% and the contribution of the proximity between values was 24%[57]. Hence, $f(x)$ modulates a conventional pairwise raw data distance ($d(S_1,S_2)$) according to the observed temporal correlation[56]. Consequently, $d\text{CORT}$ adjusts the degree of similarity between each pair of observations according to the temporal correlation and the proximity between values. This function was used to compare the reconstructed NPP between biogeographical regions during MIS 3 in the Iberian Peninsula. However, two different biogeographical regions could have experienced similar evolutionary trends in their NPP, even though their biota composition was different. Therefore, this analysis was complemented with a JSI to assess whether the reconstructed herbivore species composition in each palaeocommunity differed among biogeographical regions during the late MIS 3. The JSI was based on presence–absence data and was calculated as follows:

$$\text{JSI} = \frac{c}{(a + b + c)} \quad (4)$$

where $c$ is the number of shared species in both regions and $a$ and $b$ are the numbers of species that were only present in one of the biogeographical regions. Therefore, the higher the value the more similar the palaeocommunities of both regions were.

**Chronological assessment.** Pivotal to any hypothesis of Neanderthal replacement patterns by AMHs is the chronology of that population turnover. To this end, we used three different approaches to provide greater confidence in the results: BAMs, the OLE model and SPD of archaeological assemblages. As detailed below, each of these approaches provides complementary information about the MUPT.

First, we built a set of BAMs for the Mousterian, Châtelperronian and Aurignacian technocomplexes in each region during the MIS 3. As stated above, we compiled the available radiocarbon dates for Iberia between 55 and 30 kyr cal BP. However, not all dates or levels were included in the Bayesian chronology models.

Radiocarbon determinations obtained from shell remains were incorporated in the dataset (dataset 1); however, the local variation of the reservoir age was unknown from 55 to 30 kyr ʙᴘ. Because of uncertainties related to marine reservoir offsets, all BAMs that incorporated dates from marine shells were run twice: including and excluding these dates. All of the archaeological levels with cultural attribution issues or stratigraphic inconsistencies were excluded. The Supplementary Note provides a detailed description of the sites, levels and dates excluded and their justification. All BAMs were built for each technocomplex using the OxCAL4.2 software[55] and IntCal20 calibration curve[54].

Bayesian chronology models were built for each archaeological and palaeontological level. Then, the dates associated with each technocomplex were grouped within a single phase to determine each culture's regional appearance or disappearance. Our interest was not focused on the chronological duration of the Mousterian, Châtelperronian and Aurignacian cultures, but on the probability distribution function of the temporal boundaries of these cultures in each region. Thus, this chronological assessment aims to provide an updated chronological frame for Neanderthal replacement by AMHs in Iberia. For this reason, we did not differentiate between proto- and early Aurignacian cultures, since both are attributed to AMHs.

In each BAM, we inserted into the same sequence the radiocarbon dates associated with a given technocomplex within a start and end boundary to bracket each culture, which allowed us to determine the probability distribution function for the beginning and end moment of each cultural phase[6]. The resolution of all models was set at 20 years. We used a t-type outlier model with an initial 5% probability for each determination, but when more than one radiocarbon date was obtained from the same bone remain, we used an s-type outlier model and the combine function. The thermoluminescence dating likelihoods were included in the models, together with their associated 1σ uncertainty ranges. When dates with low agreement (<60%) were detected, we evaluated two alternatives: first, these outlier dates were automatically down-weighted in the iterative Markov Chain Monte Carlo runs; second, these outliers were removed from the sequence and the models were re-run. To further assess the robustness of the chronology obtained for the MUPT in each biogeographical region, we performed two additional sensitivity tests. First, each model was run four times and the outputs were compared. Second, we repeated each simulation after removing the youngest and oldest dates in each technocomplex and region. It should be noted that the time intervals obtained with the boundary function were larger than those obtained with the date function, given that the date function informs about the duration of the phase, whereas the boundary function informs about when the phase started and stopped being created. All of the codes and outcomes of these sensitivity tests are available in the Supplementary Note.

Different studies have used the same Bayesian modelling approach to assess the MUPT in different regions of Iberia and Europe. However, this approach is subject to certain assumptions and limitations. Since different sites are included in a single phase, the BAMs assume that all dates are related, continuous and dependent on each other, which may result in relatively short time intervals, thus potentially biasing the estimation of the first and last appearance of each culture.

OLE models are used to determine the extinction time of species and, in recent years, this approach has proved to be robust in the estimation of the first and last appearance of Palaeolithic cultures[58–60]. Unlike Bayesian statistical approaches, the OLE method does not penalize outlying data; therefore, the time intervals for the start and end of each culture are larger. OLE uses the first and last chronological occurrences of a given species or culture to assess how long it continued after the oldest or most recent confirmed occurrence[58]. The mathematical formulation of the model is available elsewhere[58]. Previous studies recommended using the five to ten oldest and youngest dates of each technocomplex to compute the first and last appearance, respectively[58,59]. We used the median and range at 95.4% CI of each date obtained with the IntCal20 calibration curve[54] as input. To address the uncertainty of the chronometric determinations, each date from within each of the ten associated date ranges was randomly drawn from a normal distribution and this was used instead of the calibrated median dates[59]. Such a randomly generated set of ages was assessed using the OLE method and the whole procedure was repeated 10,000 times[58]. In the Thermomediterranean region, the Châtelperronian is represented by only one archaeological assemblage (for details, see Supplementary Note 1). Hence, the use of the BAMs is more appropriate in this case than the OLE approach.

OLE is a well-established methodology for estimating the first and last appearance of species, but this procedure shares some limitations with the BAMs. OLE assumes a continuation of the cultures after or before the latest or earliest currently known occurrences[58]. This assumption is not valid, for example, in cases of rapid extinction, such as those derived from catastrophic events or large-scale migrations. Moreover, as in the BAMs, the OLE method does not reflect discontinuities in the fossil record. To overcome these shortcomings and to provide further support to the chronological aspects of this study, we also carried out an SPD analysis.

SPD is commonly used as a demographic proxy in Palaeolithic research. However, this method is subjected to certain assumptions[52,61]. SPD analyses assume that population size has a positive correlation with the number of dates, sites or assemblages, and that the intensity of research and preservation of archaeological sites is nearly uniform across the region of study. These assumptions

are not commonly met, so some filtering steps are necessary to ensure that SPD is a robust method with which to infer population dynamics. Following the recommendations of previous research[61], several filtering processes were used. First, dates with large error ranges were removed by eliminating all chronometric determinations with a coefficient of variation ≥0.05 (ref. [62]). Second, to ensure that each occupational unit was not overrepresented, SPD estimations were based on the chronological distribution of each archaeological assemblage, and dates from the same archaeological level were merged using the R_Combine function from OxCal before calibration[63]. Third, radiocarbon determinations obtained from shell remains were excluded because of the uncertainties related to the marine reservoir offsets, and the remaining dates were calibrated with the IntCal20 calibration curve[54]. These filtering steps overcome some of the limitations of this method, but caveats are still necessary[61]. Therefore, outcomes obtained from the SPD have not been used to inform population size in this study. Instead, they have been used as an additional method to assess the duration and particularly the frequency of occupation of each culture.

**Palaeoclimate reconstructions and validation.** The model used to estimate NPP in the current study required the following climatological input data: monthly temperature (C°), precipitation (mm per month), incoming shortwave radiation (Wm⁻²) and rainy days (days per month). These palaeoclimate parameters were obtained from the HadCM3B-M2.1 coupled general circulation model with active atmosphere, ocean and sea ice components[64]. This dataset extends back 60,000 years from 0 ʙᴘ at 0.5° resolution on a monthly timestep in the Northern Hemisphere[64]. Despite the high temporal resolution of this model, biases are frequently found when comparing global palaeoclimate models with local or regional palaeoclimate reconstructions[65]. Therefore, we assessed the accuracy of the HadC3B-M2.1 simulations in the regions of interest and performed a bias correction before using this dataset in further palaeoecological modelling.

Assessing the accuracy of palaeoclimate models is challenging due to the low availability of observational datasets[64]. Pollen assemblages constitute one of the most common palaeoenvironmental proxies used to evaluate the accuracy of palaeoclimate simulations[64,65]. Several empirical palaeoclimate reconstructions are available for the Last Glacial Maximum and the mid-Holocene, but they are less common for MIS 3. Following previous studies, pollen-based palaeoclimate reconstructions were made with weighted averaging regressions[66]. Weighted averaging regressions consider that plant species were more abundant near the climatic conditions most adapted to[67]. The predictive functions used to compute the mean annual temperature (MAT) and mean annual precipitation (MAP) from the palynological record were derived from a training set of modern pollen taxa obtained from the Eurasian Modern Pollen Database version 2 (ref. [68]). This database contains pollen taxa recovered from more than 2,000 European and Asian localities and associated climatological conditions (Supplementary Fig. 4a). This training set was used to obtain temperature and precipitation transfer functions based on pollen subsets using weighted averaging regression techniques. Prediction errors were simulated by bootstrap crossvalidation (number of boot cycles = 500). Then, the transfer functions were applied to the fossil pollen recovered from 93 palynological assemblages from 51 dated archaeological levels that cover all of the biogeographical regions of Iberia (Supplementary Fig. 4b). The percentage of each pollen taxa was obtained from the species count whenever possible and from the published palynological diagrams in most cases (dataset 2).

Before estimating the MAT and MAP from the palynological record, some filtering criteria were applied to the dataset. First, only the palynological assemblages obtained from levels dated between 55 and 30 kyr cal ʙᴘ were retained for analyses. Second, palynological assemblages with <100 pollens recovered and species with low representation (<5%) were excluded. Since terrestrial and aquatic taxa are unevenly affected by climatic conditions, it is recommended to also exclude all non-terrestrial pollen taxa, including ferns and non-pollen palynomorphs[67]. After these modifications, the percentages of each taxon were readjusted and we ensured that all of the pollen taxa recovered from the archaeological record were in the training dataset of extant pollen species. After the crossvalidation process, the correlation coefficient ($r^2$) used to estimate the MAT from the palynological record was 0.77 and the root-mean-square error (RMSE) was 4.14. However, the correlation coefficient was lower ($r^2 = 0.50$) and the RMSE higher (RMSE = 294.4) for the MAP (Supplementary Table 3). It is important to bear in mind that the palaeoclimatic reconstructions performed from the palynological record were not used in further palaeoecological reconstructions (for example, NPP estimations) but to check the predictions made from the HadCM3B-M2.1 coupled general circulation model and to assess whether bias corrections were necessary.

Different correction techniques have been proposed to rectify the observed biases in climate simulations, but the delta method has been shown to be the most robust approach to correct palaeoclimate simulations[65]. According to the delta method, the bias in a specific area is obtained from the difference between present-day observed and simulated values. Therefore, bias-corrected temperature (T) in a particular area (x) at some time (t) in the past is estimated as follows:

$$T_{sim}^{DM}(x, t) = T(x, 0) + (T_{sim}^{raw}(x, t) - T_{sim}^{raw}(x, 0))$$
$$\rightarrow T_{sim}^{raw}(x, t) + (T_{obs}(x, 0) - T_{sim}^{raw}(x, 0))$$

(5)

where $T_{sim}$ is the bias-corrected temperature in a particular area ($x$) during a specific moment in the past ($t$) according to the difference between the observed ($T_{obs}$) and predicted ($T_{sim}$) temperature for the present day.

Likewise, the bias-corrected precipitation ($P_{sim}$) for the past is estimated as follows:

$$P_{sim}^{DM}(x,t) = P_{obs}(x,0) \times \frac{P_{sim}^{raw}(x,t)}{P_{sim}^{raw}(x,0)} \rightarrow P_{sim}^{raw}(x,t) \times \frac{p_{obs}(x,0)}{P_{sim}^{raw}(x,0)} \qquad (6)$$

To assess the suitability of this bias correction method, the estimated values of MAT and MAP from the palynological record were compared with MAT and MAP obtained from the HadCM3B-M2.1 coupled general circulation model before and after using the delta correction method. Present-day values of MAT and MAP were obtained from the Climate Research Unit version 4 dataset[69]. The difference between the observed and predicted values of MAT during MIS 3 in the archaeological sites with pollen samples was 1.44 °C on average before using the bias correction method and 0.43 °C on average after using the delta correction method (Supplementary Fig. 5). The mean difference between the observed and predicted values of MAP was 1.74 mm per month before using the delta correction method and 0.82 mm per month after performing the bias correction. Accordingly, differences between the observed and predicted values were, on average, closer to zero after completing this bias correction (Supplementary Fig. 5). Moreover, there was a positive correlation between the observed and predicted values of MAT ($P<0.001$; $r=0.68$) and MAP ($P<0.001$; $r=0.67$) during MIS 3 (Supplementary Fig. 6). Accordingly, the bias-corrected values of temperatures and precipitations are in good agreement with the empirical reconstructions. These outcomes show the suitability of the delta correction method and the good correspondence between the observed and predicted rainfall and temperature values obtained from the HadCM3B-M2.1 coupled general circulation model[64] once the bias correction procedure had been performed.

**Net primary productivity.** NPP was estimated with the Lund–Potsdam–Jena General Ecosystem Simulator (LPJ-GUESS) version 4.0. model. LPJ-GUESS (https://web.nateko.lu.se/lpj-guess/) is a process-based dynamic vegetation model that simulates the structure and composition of the land cover in terms of plant functional types (PFTs)[70]. Each PFT is characterized by specific bioclimatic niche, growth form, leaf phenology, photosynthetic pathway and life history traits. The population dynamics of each PFT are determined by the competition for light, space and soil resources in each of a number of replicate patches for each simulated grid cell[70]. There are different versions of LPJ-GUESS[47] and various PFTs can be incorporated into the model[71]. In the current study, we used the standard global PFT set described by Smith et al.[70]. The model simulates the vegetation dynamics in multiple grid cells or patches to incorporate variation due to stochastic processes[70]; however, these patches are simulated independently of each other, whereby the outcomes obtained do not change if the model is run in a specific patch or in many adjacent patches[70]. Following Allen et al.[71], the model was run without the nitrogen cycle nor nitrogen limitation because nitrogen deposition is unknown for the Pleistocene. For each grid cell, 100 replicate patches with an area of 0.1 ha were simulated. For details on the global parameters used, see Supplementary Table 4.

In the present study, LPJ-GUESS was run in cohort mode to estimate NPP in the surrounding area of each archaeological and palaeontological site providing information about herbivore guild composition and human subsistence strategies during the MUPT in Iberia. The NPP accrued was allocated to leaves, roots and woody PFTs according to the specific allometric relationships of each PFT, resulting in height, diameter and biomass growth[70]. NPP was estimated at the end of each simulated year, but the model operates on a daily timestep and requires daily climate conditions. Monthly palaeoclimate drivers can be provided as input, and the model interpolates daily values from monthly values. Thus, the input climate variables for LPJ-GUESS were monthly temperature (C°), precipitation (mm per month), incoming shortwave radiation (Wm⁻²), and rainy days (days per month) for each time slice. These input climate data were obtained from the HadCM3B-M2.1 coupled general circulation model[64] after performing the delta bias correction procedure outlined above. The model also needs, as input data, the atmospheric carbon dioxide concentration (ppm) and the specific soil classes for each grid cell to incorporate texture-related variables affecting the hydrology and thermal diffusivity of the soils[70]. $CO_2$ values were obtained from Lüthi et al.[72] and soil types were obtained from Zobler[73].

All simulations were initialized with bare ground conditions (no biomass) and the model was spun up for 500 years until the simulated vegetation was in approximate equilibrium. This spin-up phase used monthly temperature, precipitation, incoming shortwave radiation and rainy days between 55 and 54.5 kyr BP[64]. Thereafter, the model was run at a monthly resolution spanning the period between 54.5 and 30 kyr BP. The accuracy and robustness of the NPP estimations were assessed in two steps. First, we compared the observed and predicted values of NPP for the present day in the regions of interest. Second, we evaluated the sensitivity of the NPP estimations to the uncertainties in the palaeoclimate variables.

Regarding the simulations of NPP for the present day, we used the same modelling protocol described above, but the input data for the spin-up phase consisted of repeated sets of monthly temperature, precipitation, incoming

shortwave radiation and rainy days data from between AD 1901 and 1930 obtained from the Climate Research Unit version 4 dataset[69]. After that, the model simulated NPP for the period between AD 1982 and 1998 to allow a more straightforward comparison with observational datasets obtained from Imhoff et al.[19]. The results obtained show a significant positive correlation between the observed and predicted values of NPP ($P<0.001$; $r=0.78$) and the mean average error was 0.071 (Supplementary Fig. 7). These results provide confidence about the performance of this dynamic vegetation model and show its suitability for simulating NPP in the regions of interest.

As mentioned previously, the temperature and precipitation values obtained from the HadCM3B-M2.1 coupled general circulation model were in good agreement with the local empirical reconstructions made with the palynological record. However, it was still necessary to assess the extent the uncertainties in the palaeoclimate simulations could affect the estimated NPP. To this end, we additionally estimated NPP from the temperatures and precipitations obtained from two alternative general circulation models at 0.5° spatial resolution[74,75]. Running the LPJ-GUESS to estimate NPP for 25,000 years (from 55,000–30,000 years BP) was a computationally intense task, so this sensitivity test focused on five specific stadial and interstadial phases: GI-12, GS-12, GS-9, GI-8 and GS-5. We selected these specific stadial and interstadial phases because they coincide with the end of the Mousterian in different regions of the Iberian Peninsula and because they represent moments of important fluctuations in NPP. The results obtained showed that the estimated NPP can vary up to 0.06 kg km⁻² yr⁻¹ on average in the Eurosiberian region, 0.07 kg km⁻² yr⁻¹ on average in the Mesomediterranean region, 0.05 kg km⁻² yr⁻¹ in the Supramediterranean region and 0.05 kg km⁻² yr⁻¹ in the Thermomediterranean region when the input palaeoclimate data are obtained from alternative general circulation models (Supplementary Table 5). However, it is important to note that the magnitude of NPP fluctuations between stadial and interstadial periods, as well as the regional differences in the estimated NPP remained constant (Supplementary Fig. 8). Therefore, this sensitivity test reinforces the robustness of the reconstructed temporal trends of NPP and the estimated differences in NPP between biogeographical regions.

**Carrying capacity of herbivores.** To date, numerous models have related the trophic structure of different ecosystems to their NPP. These studies were built on the empirical evidence for the relationship between NPP and herbivore abundance across diverse terrestrial ecosystems[76–79] (Supplementary Fig. 9). We obtained a predictive equation to estimate the total herbivore biomass (THB) that could be sustained by a given ecosystem with the data provided by different studies that cover a broad range of terrestrial ecosystems (dataset 3). An MM-type estimator for linear models was used to obtain the predictive function and to compute the 95% CI for the estimation. The obtained predictive function was:

$$\log_{10}[THB] = 1.401 \times \log_{10}[NPP] - 0.642 \qquad (7)$$

where both THB and NPP are expressed in g m⁻² yr⁻¹ (Supplementary Table 6). The biomass of a herbivore population ($B$) is commonly estimated by multiplying its population density ($D$) by the mean adult body mass of the species. Accordingly, the THB of all herbivore species in a given ecosystem could also be expressed as follows:

$$THB = \sum_{i=1}^{n} D_i \times W_i \qquad (8)$$

where $n$ is the number of herbivore species in the community, $D_i$ is the population density of species $i$ expressed in ind km⁻² and $W$ is the mean body mass of both sexes in kg. Damuth[80] demonstrated that population density changes allometrically with body size across different ecosystems:

$$D_i = c\, W_i^{-3/4} \qquad (9)$$

where $D$ is expressed in ind km⁻², $c$ is a constant and $W$ is the mean body mass in kg. Among narrow taxonomic groups, the relationship between body mass and population density may differ from the exponent of $-0.75$ (ref. [81]). Nevertheless, recent studies provide compelling evidence for the relationship of $-3/4$ on larger scales, demonstrating that, across broad taxonomic levels, the scaling factor of $-0.75$ fits the empirical observations[82]. Assuming this allometric relationship, the availability of trophic resources and the structure of regional communities, previous studies showed that herbivore abundances can be estimated[83–85]. Following these earlier works, $D$ can be substituted in equation (8):

$$THB = \sum_{i=1}^{n} (cW_i^{-3/4}) \times W_i \qquad (10)$$

Furthermore, $c$ can be estimated as follows:

$$c = \frac{THB}{\sum_{i=1}^{n} W_i^{1/4}} \qquad (11)$$

Accordingly, the biomass ($B$) of a specific herbivore population species ($i$) in a given ecosystem can be estimated with equation (12):

$$B_i = D_i \times W_i \rightarrow \left( \frac{\text{THB}}{\sum_{i=1}^{n} W_i^{1/4}} W_i^{-3/4} \right) \times W_i \tag{12}$$

To validate this model, we compiled data of 516 extant herbivore population densities in protected areas covering a wide range of different ecosystems from the TetraDENSITY database[86] and Hatton et al.[78] (dataset 3). The body mass of each herbivore species was obtained from the PHYLACINE database[53]. In 95.3% of the national parks and reserves, the estimated herbivore population densities were significantly correlated ($P < 0.05$) with the observed herbivore abundance, with correlation coefficients ($r$) ranging between 0.49 and 0.98 (Supplementary Fig. 10). The only two localities where there was no correspondence between the observed and predicted densities were tropical ecosystems with a low number of identified species because of the lack of density surveys (dataset 3). When the observed and predicted values were analysed together across all ecosystems, there was a significant positive correlation between the observed and predicted values ($P < 0.001$; $r = 0.65$) (Supplementary Fig. 11). These results confirm the validity of this macroecological modelling approach.

This approach assumes that the biomass of each herbivore population depends on the bottom-up processes of food chain regulation driven by NPP, the specific herbivore guild composition in a given ecosystem and the allometric relationships between body mass and population density. Even in glacial times, one would expect to find a positive relationship between NPP and THB, but megafauna could have different grass exploitation efficiencies than extant large herbivores, as previously suggested to explain the productivity paradox of the mammoth steppe ecosystems[23]. Moreover, this approach does not consider other factors that affect herbivore abundances, such as species migratory movements or predatory pressure. For these reasons, the herbivore biomass derived from this modelling approach should not be interpreted as an estimation of the actual herbivore biomass, but rather as the carrying capacity of each herbivore population species in a specific region. It is worth noting that the current study did not aim to estimate precisely the biomass of each herbivore population, but rather to investigate to what extent MIS 3 climate changes affected the potential biomass availability for secondary consumers.

This modelling approach was applied to each MIS 3 herbivore palaeocommunity in the biogeographical regions of Iberia. A palaeocommunity was determined by the LFAs found in a specific biogeographical region during a particular time span without significant species turnover[87]. However, different gaps may affect the number and type of herbivore species that conform to a given palaeocommunity (for example, uneven sampling effort or taphonomical biases). A method for partially correcting these drawbacks in palaeoecology is the minimum census technique. According to this method, the species composition of each palaeocommunity is not inferred from the species occurrences in each specific time interval (that is, stadial and interstadial phase), but from the chronological range of each species in each biogeographical region. However, this technique does not solve the issues related to the completeness of the species lists that conform to each palaeocommunity. To further assess the totality of the number of species in each palaeocommunity, a rarefaction analysis was performed. Rarefaction (interpolation) and prediction (extrapolation) curves are commonly used to assess species richness according to the sampling effort. This method assumes that when the number of samples increases, the species richness approaches an asymptote. Accordingly, the observed species richness was compared with the expected richness in a sample size of 100 LFAs in each palaeocommunity and a bootstrap method was applied ($n = 500$) to obtain the 95% CI for each diversity estimate. The outcomes obtained show that the actual species richness in each palaeocommunity was close to the asymptote (Supplementary Fig. 12) and, consequently, increasing the sample size of LFAs would not increase the number of species in each palaeocommunity (Supplementary Table 7) significantly.

**Reporting summary.** Further information on research design is available in the Nature Research Reporting Summary linked to this article.

## Data availability

Datasets 1, 2 and 3 are available from GitHub (https://github.com/ERC-Subsilience/Data-and-code-associated-with-Iberia-Neanderthal-ecosystems-productivity_Nature-Ecology-Evolution) and Zenodo (https://doi.org/10.5281/zenodo.6832689).

## Code availability

The R codes used to perform all of the analyses reported in this manuscript are available from GitHub (https://github.com/ERC-Subsilience/Data-and-code-associated-with-Iberia-Neanderthal-ecosystems-productivity_Nature-Ecology-Evolution) and Zenodo (https://doi.org/10.5281/zenodo.6832689). The source code for LPJ-GUESS version 4.0 can be obtained on request through Lund University (http://web.nateko.lu.se/lpj-guess).

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

## Acknowledgements

This research was funded by the European Research Council under the European Union's Horizon 2020 Research and Innovation Programme (grant agreement number 818299;

SUBSILIENCE project; https://www.subsilience.eu). We thank all of our colleagues from the EvoAdapta group for constant enriching discussions.

## Author contributions

A.B.M.-A. and M.V.-C. designed the study. M.V.-C., D.O. and T.H. contributed to the model development. M.V.-C. and D.O. analysed the data. M.V.-C., A.B.M.-A. and D.O. contributed to evaluating the outcomes M.V.-C. led the writing with critical input from A.B.M.-A., D.O. and T.H.

## Competing interests

The authors declare no competing interests.

## Additional information

**Correspondence and requests for materials** should be addressed to M. Vidal-Cordasco or A. B. Marín-Arroyo.

# Reporting Summary

## Statistics

For all statistical analyses, confirm that the following items are present in the figure legend, table legend, main text, or Methods section.

| n/a | Confirmed | |
|---|---|---|
| ☒ | ☐ | The exact sample size (*n*) for each experimental group/condition, given as a discrete number and unit of measurement |
| ☒ | ☐ | A statement on whether measurements were taken from distinct samples or whether the same sample was measured repeatedly |
| ☐ | ☒ | The statistical test(s) used AND whether they are one- or two-sided<br>*Only common tests should be described solely by name; describe more complex techniques in the Methods section.* |
| ☐ | ☒ | A description of all covariates tested |
| ☐ | ☒ | A description of any assumptions or corrections, such as tests of normality and adjustment for multiple comparisons |
| ☐ | ☒ | A full description of the statistical parameters including central tendency (e.g. means) or other basic estimates (e.g. regression coefficient) AND variation (e.g. standard deviation) or associated estimates of uncertainty (e.g. confidence intervals) |
| ☐ | ☒ | For null hypothesis testing, the test statistic (e.g. $F$, $t$, $r$) with confidence intervals, effect sizes, degrees of freedom and $P$ value noted<br>*Give P values as exact values whenever suitable.* |
| ☒ | ☐ | For Bayesian analysis, information on the choice of priors and Markov chain Monte Carlo settings |
| ☒ | ☐ | For hierarchical and complex designs, identification of the appropriate level for tests and full reporting of outcomes |
| ☐ | ☒ | Estimates of effect sizes (e.g. Cohen's *d*, Pearson's *r*), indicating how they were calculated |

*Our web collection on statistics for biologists contains articles on many of the points above.*

## Software and code

Policy information about availability of computer code

| | |
|---|---|
| Data collection | The LPJ-GUESS v4.0 was used to estimate the Net Primary Productivity. LPJ-GUESS can be obtained on request through Lund University (http://web.nateko.lu.se/lpj-guess) |
| Data analysis | The R codes used to perform all the analyses reported in the manuscript are available on Github: https://github.com/ERC-Subsilience/Data-and-code-associated-with-Iberia-Neanderthal-ecosystems-productivity_Nature-Ecology-Evolution; Vidal-Cordasco, M., Ocio, D., Hickler, T., & Marín-Arroyo, A. B. (2022). ERC-Subsilience/Data-and-code-associated-with-Iberia-Neanderthal-ecosystems-productivity_Nature-Ecology-Evolution: Data-and-code-associated-with-Iberia-Neanderthal-ecosystems-productivity_Nature-Ecology-Evolution. https://doi.org/10.5281/zenodo.6826921) |

For manuscripts utilizing custom algorithms or software that are central to the research but not yet described in published literature, software must be made available to editors and reviewers. We strongly encourage code deposition in a community repository (e.g. GitHub). See the Nature Portfolio guidelines for submitting code & software for further information.

## Data

Policy information about availability of data

All manuscripts must include a data availability statement. This statement should provide the following information, where applicable:

- Accession codes, unique identifiers, or web links for publicly available datasets
- A description of any restrictions on data availability
- For clinical datasets or third party data, please ensure that the statement adheres to our policy

All data generated or analysed during this study are available on Github: https://github.com/ERC-Subsilience/Data-and-code-associated-with-Iberia-Neanderthal-ecosystems-productivity_Nature-Ecology-Evolution; Vidal-Cordasco, M., Ocio, D., Hickler, T., & Marín-Arroyo, A. B. (2022). ERC-Subsilience/Data-and-code-

# Field-specific reporting

Please select the one below that is the best fit for your research. If you are not sure, read the appropriate sections before making your selection.

☐ Life sciences ☐ Behavioural & social sciences ☒ Ecological, evolutionary & environmental sciences

For a reference copy of the document with all sections, see nature.com/documents/nr-reporting-summary-flat.pdf

# Ecological, evolutionary & environmental sciences study design

All studies must disclose on these points even when the disclosure is negative.

| | |
|---|---|
| Study description | We test whether the temporal and spatial replacement patterns of H. neanderthalensis by H. sapiens in Iberia were affected by alterations in the ecosystem productivity.To this end, we integrated three modelling approaches. First, we built Bayesian age models for each cultural techno-complex in the four biogeographic regions of Iberia. This analysis was performed with the OxCAL4.2 software and the INTCAL20 calibration curve. This chronological assessment was complemented with Optimal Linear Estimation and Summed Calibrated Distribution of dated archaeological assemblages. Second, we used a generalised dynamic vegetation model (LPJ-GUESS v.4) with tested climate inputs from an atmospheric general circulation model to estimate the evolution of Net Primary Productivity (NPP) between 55 and 30 ky BP in each archaeological and paleontological site of the MIS 3. A sensitivity analysis was performed to assess the robustness of the NPP estimations by using the climate inputs obtained from alternative paleoclimatic models. Lastly, we validated a macroecological model against empirical present-day herbivore densities from a broad range of terrestrial ecosystems and used this modelling approach to estimate the herbivore carrying capacity in each stadial and interstadial phase. |
| Research sample | This study includes the chronometric dates and the herbivore species recovered from 62 archaeological and 7 paleontological sites dated in the MIS 3. We used the climate datasets from the CRU v.4 and the HadCM3B-M2.1 coupled general circulation model to estimate the Net Primary Productivity. The Eurasian Modern Pollen Database v.2 was used to obtain temperature and precipitation transfer functions based on pollen subsets. These predictive functions were applied to the fossil pollen recovered from 93 palynological assemblages from 51 dated archaeological levels from all the biogeographic regions included in this study. The Phylacine dataset was used to obtain the body mass of the herbivore species included in the study. To validate the macroecological model that estimates herbivore abundances, we used data of 516 extant herbivore population densities obtained from the TetraDENSITY database. |
| Sampling strategy | To our knowledge, this study includes all the archaeological sites of the Middle to Upper Palaeolithic transition in Iberia. A rarefaction test was used to assess the sample size used to reconstruct the paleocommunity composition in each biogeographic region. All data used in this study was compiled from the literature or obtained from the modeling approaches described in the manuscript. |
| Data collection | Data was collected by Vidal-Cordasco & Marín-Arroyo from the literature. |
| Timing and spatial scale | Archaeo-paleontological data was collected from the literature between 01/04/2021to 01/11/2021. Data from the LPJ-GUESS model was obtained from 01/06/2021 to 01/11/2021. Regarding the spatial scale, data obtained focused in the Iberian Peninsula. |
| Data exclusions | We excluded from the age models some archaeological levels with cultural or stratigraphic inconsistencies. Some of these exclusion criteria were pre-established (e.g. only included archaeological levels with Mousterian, Châtelperronian or Aurignacian remains were included); however, after performing the sensitivity tests, we decided to exclude all dates obtained from marine shell remains due to the uncertainties with the reservoir effects. The specific rationale behind each exclusion is discussed in Supplementary Note. |
| Reproducibility | To verify the findings, all analyses and models were re-run and different sensitivity analyses were performed. The results reported in the manuscript are reproducible with the data and codes available on Github: https://github.com/Marco-Vidal/Data-and-codes-associated-with-Vidal-Cordasco-et-al.-Nature-Ecology-Evolution-. All attempts to reproduce the experiments were successful. |
| Randomization | Samples were allocated into groups according to: 1) the biogeographic region of each archaeological or paleontological site, 2) the chronological and cultural attributions of each archaeological and paleontological unit, 3) the evolutionary trends of the Net Primary Productivity, and 4) the herbivore guild composition. |
| Blinding | Blinding was not relevant to this study since the validity of the results are independent of the individuals involved in the study. |

Did the study involve field work? ☐ Yes ☒ No

# Reporting for specific materials, systems and methods

We require information from authors about some types of materials, experimental systems and methods used in many studies. Here, indicate whether each material, system or method listed is relevant to your study. If you are not sure if a list item applies to your research, read the appropriate section before selecting a response.

## Materials & experimental systems

| n/a | Involved in the study |
|---|---|
| ☒ | ☐ Antibodies |
| ☒ | ☐ Eukaryotic cell lines |
| ☐ | ☒ Palaeontology and archaeology |
| ☒ | ☐ Animals and other organisms |
| ☒ | ☐ Human research participants |
| ☒ | ☐ Clinical data |
| ☒ | ☐ Dual use research of concern |

## Methods

| n/a | Involved in the study |
|---|---|
| ☒ | ☐ ChIP-seq |
| ☒ | ☐ Flow cytometry |
| ☒ | ☐ MRI-based neuroimaging |

# Palaeontology and Archaeology

| | |
|---|---|
| Specimen provenance | All archaeological material in this study were already published in different research papers |
| Specimen deposition | *Indicate where the specimens have been deposited to permit free access by other researchers.* |
| Dating methods | Any new date is provided in this study |

☒ Tick this box to confirm that the raw and calibrated dates are available in the paper or in Supplementary Information.

| | |
|---|---|
| Ethics oversight | No ethical approval was required |

Note that full information on the approval of the study protocol must also be provided in the manuscript.

