## [Peer Review File · Nature Ecology & Evolution]

Peer Review Information

Journal: Nature Ecology & Evolution

Manuscript Title: Ecosystem productivity affected the spatiotemporal disappearance of Neanderthals in Iberia

Corresponding author name(s): A. Marín-Arroyo, M. Vidal-Cordasco

Editorial Notes:

Reviewer Comments & Decisions:

Decision Letter, initial version:
--

4th April 2022

Dear Dr Marín-Arroyo,

Your Article, "Ecosystems productivity affected the spatiotemporal disappearance of Neanderthals in Iberia" has now been seen by three reviewers. You will see from their comments copied below that while they find your work of considerable potential interest, they have raised quite substantial concerns that must be addressed. In light of these comments, we cannot accept the manuscript for publication, but would be very interested in considering a revised version that addresses these serious concerns.

We hope you will find the reviewers' comments useful as you decide how to proceed. If you wish to submit a substantially revised manuscript, please bear in mind that we will be reluctant to approach the reviewers again in the absence of major revisions.

We are particularly concerned about the reviewers' comments on Bayesian modelling of dates (raised by two reviewers), and ask you to pay particular attention to reviewer 1's recommendations for alternative methods of estimating first and last appearance dates.

If you choose to revise your manuscript taking into account all reviewer and editor comments, please highlight all changes in the manuscript text file [OPTIONAL: in Microsoft Word format].

We are committed to providing a fair and constructive peer-review process. Please do not hesitate to

contact us if there are specific requests from the reviewers that you believe are technically impossible or unlikely to yield a meaningful outcome.

* Include a "Response to reviewers" document detailing, point-by-point, how you addressed each referee comment. If no action was taken to address a point, you must provide a compelling argument. This response will be sent back to the referees along with the revised manuscript.

* If you have not done so already we suggest that you begin to revise your manuscript so that it conforms to our Article format instructions at <http://www.nature.com/natecolevol/info/final-submission>. Refer also to any guidelines provided in this letter.

[REDACTED]

If you wish to submit a suitably revised manuscript we would hope to receive it within 6 months. If you cannot send it within this time, please let us know. We will be happy to consider your revision so long as nothing similar has been accepted for publication at Nature Ecology & Evolution or published elsewhere.

Nature Ecology & Evolution is committed to improving transparency in authorship. As part of our efforts in this direction, we are now requesting that all authors identified as 'corresponding author' on published papers create and link their Open Researcher and Contributor Identifier (ORCID) with their account on the Manuscript Tracking System (MTS), prior to acceptance. This applies to primary research papers only. ORCID helps the scientific community achieve unambiguous attribution of all scholarly contributions. You can create and link your ORCID from the home page of the MTS by clicking on 'Modify my Springer Nature account'. For more information please visit www.springernature.com/orcid.

Thank you for the opportunity to review your work.

[REDACTED]

Reviewer expertise:

Reviewer #1: hunter gatherer ecology and modelling

Reviewer #2: Middle Palaeolithic archaeology and ecology

Reviewer #3: Middle-Upper Palaeolithic transition

Reviewers' comments:

Reviewer #1 (Remarks to the Author):

By looking at the ecosystem productivity – and not just NPP, but also herbivore carrying capacity, this paper makes an important contribution towards understanding the causes of Neanderthal extinction as well as more general understanding of cultural and ecological changes. You must be applauded for having done great (and huge) work with evaluating the climate model and combining data for and building models of herbivore carrying capacity.

I can imagine some archaeologists and paleontologists whining that this “just” a modeling study, because you rely heavily on mechanistic climate and vegetation models and statistical (or other) models calibrated with modern data (pollen-based climate reconstructions, herbivore biomass models), but that is their problem. By combining insights and data from models with archaeological and other proxies we really can start to build understanding about the past.

Below, I have highlighted two partly interrelated issues for you to consider.

1. My main concern relates to the use of Bayesian age modeling to detect the beginning and end of archaeological cultures, which features as an important component in your analysis. Please, forgive my rather long rant here.

I believe, or at least hope, that the developers of OxCal did not intend Bayesian age modeling to be used the way that has become fairly common among archaeologists including your paper. By inserting the dates of a cultural phase (e.g. Aurignacian) coming from a number of different sites into a Phase model means that OxCal tries to force them to form a single “phase”. Given this prior information (“these dates from a phase”), the algorithm basically tries to see if it is possible to juggle with individual probability distributions so that they become abutting, i.e., forming a continuous phase. As a result of modeling, all the dates are squeezed together, which means that the modeled age range of a culture/techno complex becomes reduced as compared to the original non-modeled age range, sometimes quite considerably – the beginning of the phase becomes younger and the end older.

This kind of application of Bayesian age modeling of cultural phases, while fairly common, probably violates the assumptions of OxCal Phase-modeling and at least completely ignores the established

3understanding concerning the formation of archaeological and fossil records. As a result of interplay of sampling and time-dependent loss of archaeological and fossil material, it is highly unlikely that we will ever find the first or last instance of a cultural trait and therefore temporal ranges in archaeological and fossil records are always shorter than true ranges (e.g. Perreault, 2019, p. 101). Application of Bayesian age models to age ranges of cultures significantly propagates this problem by further shrinking the temporal ranges.

Bayesian age models might help to deal with uncertainties related to radiocarbon measurements, but clearly not with uncertainties related to formation of the record (including sampling). Even with uncertainties related to radiocarbon measurements, Bayesian age models work only as long as the prior information given to the model is valid. However, there are absolutely no justifications for the assumption that e.g. Aurignacian radiocarbon date record would be without holes created by formation processes, incomplete sampling or true absences of humans in research area during some short intervals. Yet, in practice, OxCal Phase-modeling seems to require this kind of "perfect" data, where true ages would "touch" each other forming a neat phase.

By applying Bayesian age model to incomplete data, typical of archaeology and paleontology will create distorted views of the temporal ranges of cultures or species (typically too short). Therefore, people should stop using Bayesian age models to estimate first (FAD) or last appearance (LAD) dates of archaeological cultures and I urge you to do so as well. Instead, one could try to apply confidence intervals around FAD or LAD in the non-modeled data (e.g. Bobe and Wood, 2022) before relating archaeological and environmental data, and/or try to estimate "true" FAD or LAD using methods such as optimal linear estimation (Key et al., 2021). At least, you could estimate the time ranges and, consequently, FAD and LAD with non-modeled dates using, e.g., summed probability distributions or kernel density estimation to see how the results would change. Because this approach would provide more robust and less biased results than phase-modeling (we know that in this kind of applications phase modeling likely produces too short time ranges), I suggest that you would include results based on non-modeled dates in the main results of your paper. However, if you want to stick with Bayesian modeling for some reason, you should include analysis based on non-modeled dates at least in the supplement to show that the use of Bayesian age modeling does not significantly influence your results. If there are no significant differences between approaches, it is not that big deal, which approach you want to show in your main results.

2. The second issue that I would like to highlight relates to the link between the end of Mousterian and drop in productivity (NPP and herbivore carrying capacity). When I look at your figure 4 and the probability distribution for the end of Mousterian in the Eurosiberian region, I can't see a link between the end of Mousterian and the drop of productivity during the GS-12. Given the probability distribution of the Mousterian end, the end likely occurred before 45ka. Although possible based on the confidence limits, it is much more unlikely that the end would have occurred after 45ka. Therefore the link between the Mousterian end and the drop in NPP does not look very strong in the figure 4. The figure 5 might also be a little misleading, because you have extended the Mousterian up to the end of GI-12 (44.28 ka) in the Eurosiberian region, although it is quite unlikely that it lasted so long according to your analyses, as explained above.

So my question is, how justifiable is the link between the end of Mousterian and the drop in productivity in the Eurosiberian region? Interestingly, in the nearby Submediterranean region, Mousterian somehow survived the dramatic drop in productivity during GS-12 and disappeared during the next stadial.

However, your result might be influenced (biased) by Bayesian age modeling and the related ignorance of archaeological formation processes that I highlighted in the comment 1. I nevertheless find it a really interesting result that Mousterian and maybe Neanderthals themselves seem to disappear earlier in the regions, where fluctuations in productivity are stronger.

References

Bobe, R., Wood, B., 2022 (in press). Estimating origination times from the early hominin fossil record. *Evol. Anthropol. Issues News Rev.* n/a. <https://doi.org/10.1002/evan.21928>

Key, A., Roberts, D., Jarić, I., 2021. Reconstructing the full temporal range of archaeological phenomena from sparse data. *J. Archaeol. Sci.* 135, 105479. <https://doi.org/10.1016/j.jas.2021.105479>

Perreault, C., 2019. *The Quality of the Archaeological Record*. The University of Chicago Press, Chicago.

Reviewer #2 (Remarks to the Author):

This article uses diverse data to evaluate the hypothesis that fluctuations in biomass availability affected the differential disappearance of Neanderthal populations in the Iberian Peninsula. The aspects of the paper that touch on calculating NPP and carrying capacity are outside my direct areas of expertise to the editor will need to rely on the comments of other reviewers when it comes to those aspects.

While I find this paper to be innovative in that it is examining the question of Neanderthal disappearance in Iberia with modified approaches compared to what has been done in the past (e.g. d'Errico and Sanchez Goni 2003 QSR; Banks et al. 2008, which curiously is not cited since they clearly showed that climatic fluctuations were not the main driver behind Neanderthal disappearance), I find that it falls a bit short in laying out its chronological foundation, a foundation which is all important for this type of study and for which we have much improved methods today.

First, I take the current correlations between the end/beginning of an archaeological phase (e.g., end of Mousterian) in a particular region and a particular Greenland Interstadial or Stadial with a bit of caution and this for two reasons.

First is the reliance on OxCal. We know that OxCal artificially compresses chronological intervals (Posterior Distribution Functions) based on how it treats dates contained within a defined phase. This is based on its use of the NSBC prior and the fact that it places two hyperparameters at the beginning and end of each phase and the effect of this is that it tends to pull the PDF around the date in a phase

5that has the smallest standard error (Lanos and Philippe 2018, Communications for Statistical Applications and Methods)—this is especially true for ages that have large standard errors and is visible in almost any study on the late MP or early UP that employs OxCal. The software used by Lanos and Philippe, named ChronoModel, does not use the NSBC prior and only places hyperparameters at the beginning and end of the entire age model, such that it better manages chronological data and their distribution within each phase. Therefore, I would suggest that the authors test their OxCal model against one produced with ChronoModel to test their initial results and evaluate whether they are working with a chronology that is artificially precise. It is freely available at <https://chronomodel.com> and one can also find references to a number of articles that have employed it, as well as to those that describe its mathematics.

The second issue associated with correlating the Iberian record to climatic chronologies is related to that of radiocarbon calibration and making links to Greenland calendar year paleoclimatic records. One article that details this problem is Giaccio et al. (2017, Sci. Reports) and their examination of how IntCal09 and 13 underestimated real ages, something that is relatively resolved with IntCal20 (Bard et al, 2020 PNAS). This issue, though, is not addressed in the article, so it would be good to see that corrected in order to add weight to the foundational chronological aspects of this study. Finally, it would be good to see a discussion of why the correlations between archaeological intervals or transitions and Greenland climatic chronology are thought to be robust, especially considering the fact that the maximum counting errors for the Greenland record are not anecdotal (ca. 1500-1700 years for the late MP and initial UP; Rasmussen et al. 2014, QSR).

I do not have an issue with the NPP and biomass aspects of the paper, but if one is to have any confidence in the conclusions that are drawn from the results, we must be entirely convinced that our conclusions are drawn from correct associations between the archaeological record and the corresponding paleoclimatic context, associations that I think merit being more solidly demonstrated.

Reviewer #3 (Remarks to the Author):

This paper assesses the effects of ecosystem productivity (Net Primary Productivity and Total Herbivore Biomass) on extinction of the Neanderthals in four biogeographic regions of Iberia. It is a complex and highly robust paper that combines Bayesian age modelling of the major industries, palaeoclimatic reconstructions, and resultant estimates of NPP and herbivore biomass. Importantly, the paper does not suggest that changes in NPP and herbivore biomass would have effected the two species differently - it simply combines age estimates for start and end dates of the various industries to examine alignments between environmental changes and the decline of Neanderthal (and AMH) populations. The fact that results for the four regions are quite different is of interest, and demonstrates an objective approach (i.e. the authors genuinely set out to test their hypothesis rather than to prove it, which is of course the way it should be).

I have no major problems with the paper, but below are a series of questions / suggestions for clarification:

6Line 86 (etc.) You refer to 'moments' - maybe 'periods' or 'phases' would be better.

Line 159ff. The Eurosiberian region is the only one in which climate seems to play a key role - it would be useful to explicitly mark the decrease in resource availability on Figure 3.

Line 244-281. The caveats and limitations are made very clear here, which adds to the overall robusticity of the paper.

I couldn't find anything about how radiocarbon dates are treated when they come from the same site (or same stratigraphic layer). Presumably multiple dates on the same stratigraphic layer are averaged in some way before being used in the Bayesian models, otherwise there is a risk that sites that have had extensive data programmes are over-represented (in other words, the results end up being skewed by research effort). Could you discuss this briefly (preferably in the main 'Methods' text).

Equations 1 & 2: What are S_1 , S_2 , p , u , and v ? Presumably S_1 and S_2 are time series 1 and 2, so is p the length of the time series, with u and v values from each time series? I'm guessing that this is what is happening, but it would be better not to have to guess. Likewise, what is a 'raw-data distance'? Is this a Euclidean distance? Is it total distance between the two whole series, or for a series of pairwise points?

Equation 3: What value of the parameter k is actually used? How and why was this value chosen?

Equations 5 and 6 have very different forms, yet they are delta corrections of temperature and precipitation respectively. Why do they have such different forms?

Finally, a couple of points that require clarification in terms of the faunal evidenc. What if it is not overall THB that is important, but the abundances of particular species (and what if this differs between Neanderthals and AMH)? Since you have the data, why not consider this in relation to faunal assemblages from Neanderthal and AMH sites? Are there any differences? Are any species particularly over-represented?

Are you only using archaeological faunal assemblages as indicators of the available faunal species at the time? This approach is useful in that it directly reflects subsistence practices, but of course it is not a reflection of overall faunal abundances in the region (because there is likely to be some selectivity in hunting strategies, which biases species representation in archaeological assemblages). It would be good to see this issue discussed (even briefly) in the main Methods text.

Author Rebuttal to Initial comments

The changes introduced in the Main text are highlighted in blue colour. Below you will find a point-to-point reply to all the reviewers' comments. First, we copy the Reviewers' comments in bold, and then we describe how we have addressed each of the issues:

Reviewer #1 (Remarks to the Author):

1) By looking at the ecosystem productivity – and not just NPP, but also herbivore carrying capacity, this paper makes an important contribution towards understanding the causes of Neanderthal extinction as well as more general understanding of cultural and ecological changes. You must be applauded for having done great (and huge) work with evaluating the climate model and combining data for and building models of herbivore carrying capacity. I can imagine some archaeologists and paleontologists whining that this “just” a modeling study, because you rely heavily on mechanistic climate and vegetation models and statistical (or other) models calibrated with modern data (pollen-based climate reconstructions, herbivore biomass models), but that is their problem. By combining insights and data from models with archaeological and other proxies we really can start to build understanding about the past.

R.1. We very much appreciate the positive feedback of Reviewer 1 regarding our manuscript.

Below, I have highlighted two partly interrelated issues for you to consider.

2) My main concern relates to the use of Bayesian age modeling to detect the beginning and end of archaeological cultures, which features as an important component in your analysis. Please, forgive my rather long rant here. I believe, or at least hope, that the developers of OxCal did not intend Bayesian age modeling to be used the way that has become fairly common among archaeologists including your paper. By inserting the dates of a cultural phase (e.g. Aurignacian) coming from a number of different sites into a Phase model means that OxCal tries to force them to form a single “phase”. Given this prior information (“these dates from a phase”), the algorithm basically tries to see if it is possible to juggle with individual probability distributions so that they become abutting, i.e., forming a continuous phase. As a result of modeling, all the dates are squeezed together, which means that the modeled age range of a culture/techno complex becomes reduced as compared to the original non-modeled age range, sometimes quite considerably – the beginning of the phase becomes younger and the end older. This kind of application of Bayesian age modeling of cultural phases, while fairly common, probably violates the assumptions of OxCal Phase-modeling and at least completely ignores the established understanding concerning the formation of archaeological and fossil records. As a result of interplay of sampling and time-dependent loss of archaeological and fossil material, it is highly unlikely that we will ever find the first or last instance of a cultural trait and therefore temporal ranges in archaeological and fossil records are always shorter than true ranges (e.g. Perreault, 2019, p. 101). Application of Bayesian age models to age ranges of cultures significantly propagates this problem by further shrinking the temporal ranges.

8Bayesian age models might help to deal with uncertainties related to radiocarbon measurements, but clearly not with uncertainties related to formation of the record (including sampling). Even with uncertainties related to radiocarbon measurements, Bayesian age models work only as long as the prior information given to the model is valid. However, there are absolutely no justifications for the assumption that e.g. Aurignacian radiocarbon date record would be without holes created by formation processes, incomplete sampling or true absences of humans in research area during some short intervals. Yet, in practice, OxCal Phase-modeling seems to require this kind of “perfect” data, where true ages would “touch” each other forming a neat phase. By applying Bayesian age model to incomplete data, typical of archaeology and paleontology will create distorted views of the temporal ranges of cultures or species (typically too short). Therefore, people should stop using Bayesian age models to estimate first (FAD) or last appearance (LAD) dates of archaeological cultures and I urge you to do so as well. Instead, one could try to apply confidence intervals around FAD or LAD in the non-modeled data (e.g. Bobe and Wood, 2022) before relating archaeological and environmental data, and/or try to estimate “true” FAD or LAD using methods such as optimal linear estimation (Key et al., 2021). At least, you could estimate the time ranges and, consequently, FAD and LAD with non-modeled dates using, e.g., summed probability distributions or kernel density estimation to see how the results would change. Because this approach would provide more robust and less biased results than phase-modeling (we know that in this kind of applications phase modeling likely produces too short time ranges), I suggest that you would include results based on non-modeled dates in the main results of your paper. However, if you want to stick with Bayesian modeling for some reason, you should include analysis based on non-modeled dates at least in the supplement to show that the use of Bayesian age modeling does not significantly influence your results. If there are no significant differences between approaches, it is not that big deal, which approach you want to show in your main results.

R.2. We agree with Reviewer 1: since different sites are included in a single “Phase” in OxCal, the Bayesian age models assume that all dates are related, continuous, and dependent from each other. Thus, Reviewer 1 is right regarding the limitations and assumptions of the Bayesian age models, which have not been acknowledged in the previous version of the manuscript. However, it should be noted that, in this study, we did not use the function “Date” in the Bayesian age models to estimate the duration of each culture, but the function “Boundary”. Effectively, as Reviewer 1 says, if different dates from different sites are included within a single phase, the duration (obtained from the “Date” function) would be artificially short, but the use of the “Boundary” function overcomes this issue as it informs about when that phase started and stopped to being created, not about its duration. Thus, the “Boundary” function provides larger intervals. In any case, we agree with Reviewer 1 regarding the limitations and assumptions of this method and appreciate the alternative approaches proposed by Reviewer 1 to assess the timing of the Middle to Upper

Palaeolithic transition in each region. Thus, we followed all the recommendations raised by Reviewer 1 and, in the new version of the manuscript, the end of the Mousterian and the beginning and end of the Aurignacian techno-complexes were estimated in each region with 1) Optimal Linear Estimations (OLE) and with 2) Summed Probability Distributions (SPD). Below, we compare the results obtained with each one of these approaches:

1. Optimal Linear Estimation (OLE).

The OLE method has proved to be robust in the estimation of the first (FA) and last (LA) appearance of Palaeolithic cultures (Bebber & Key, 2022; A. J. M. Key et al., 2021, 2021). As suggested by Reviewer 1, we followed the same procedure as used in (A. Key et al., 2021a) to estimate the first and last appearance of each culture with the OLE model. It is recommended to use the 5-10 oldest and youngest dates of each technocomplex to compute the LA and FA; using more than 10 dates does not alter the results (Bebber & Key, 2022; A. J. M. Key et al., 2021). We used the median and the range at 95.4%CI of each date obtained with the IntCal20 calibration curve as input. To address the uncertainty of the chronometric determinations, each date from within each of the 10 associated date ranges was randomly drawn from a normal distribution and used instead of the calibrated median dates. Such randomly generated set of ages was assessed with the OLE method, and the whole procedure was repeated 10,000 times (Bebber & Key, 2022; A. J. M. Key et al., 2021). These analyses were performed with the sExtinct R package and the results obtained can be replicated with the code and data available at <https://github.com/Marco-Vidal/Data-and-codes-associated-with-Vidal-Cordasco-et-al.-Nature-Ecology-Evolution-> Results are summarised in the following Table 1.

Region	Culture	Boundary	BAM		OLE			
					Central range		Resampling	
			From	to	From	to	From	to
Eurosiberian	Mousterian	End	47.0 8	44.0 6	45.2 9	43.8 8	44.9 6	43.0 6
	*Châtelperronian	Start	43.8 3	42.0 2	53.2	43.6 2	65.8 7	47.3 8
		End	42.3 4	41.0 2	41.8 8	41.0 4	41.0 3	37.6 5
	Aurignacian	Start	43.3	42.0	46.6	43.9	51.2	45.2

103	5	4	7	2	5
		End	36.0 1	34.7 3	35.3 6	35.0 2	35.2 0	34.6 1
Supramediterranean	*Mousterian	End	42.7 6	40.4 8	41.9 3	39.6 1	41.7 9	38.1 9
Mesomediterranean	Mousterian	End	41.8 8	40.8 0	41.1 6	39.8 7	41.1 4	39.8 8
	Aurignacian	Start	42.6 2	41.4 8	44.1 0	42.4 0	44.4 7	42.5 5
		End	36.9 7	35.6 9	36.2 5	35.5 4	36.1 3	35.2 4
Thermomediterranean	Mousterian	End	36.6 2	32.6 4	35.4 3	32.8 7	35.4 4	32.9 2
	*Châtelperronian	Start	42.7 6	39.2 8	44.1 4	39.7 1	45.9 5	39.8 3
		End	40.2 4	36.3 7	24.9 4	15.7 3	22.8 2	18.9 2
	Aurignacian	Start	42.4 5	41.4 4	43.5 2	40.3 1	44.5 2	40.5 8
		End	33.9 5	32.7 4	30.7	28.9 2	30.6 9	28.9 5

Table 2. Upper and lower bounds of each model's 95% CI for the start and end of the Mousterian, Châtelperronian and Aurignacian techno-complexes in each biogeographic region according to the Bayesian Age Models (BAM) and the Optimal Linear Estimation models (OLE). The asterisk (*) indicates that the sample size was lower than 10. For details, see Supplementary Note 1.

According to the OLE method, the end of the Mousterian (45.29-43.88 ka cal BP), Châtelperronian (41.88-41.04 ka cal BP) and Aurignacian (35.36-35.02 ka cal BP) in the Eurosiberian region is consistent with the Bayesian age models (Table 1). This provides support to the proposed chronology in the previous version of the manuscript for the end of the Mousterian in the

Eurosiberian region. The main difference between the outcomes obtained from the Bayesian age models and that obtained with the OLE method is the older beginning and the larger temporal ranges of the later for the start of the Châtelperronian (53.2-43.62 ka cal BP) and the start of the Aurignacian (46.64-43.97 ka cal BP), particularly after the resampling procedure (Table 1). These larger intervals at 95% CI are due to the standard errors of some chronometric determinations, particularly from dates obtained with OSL/TL, or samples whose calibrated ages are close to the limit of the radiocarbon calibration curves. For that reason, the chronological intervals for the start of each culture are substantially larger than those obtained for the end of the same cultures. As a result, the outcomes obtained from the OLE models provide support to the previous chronological assessment, but also reveal a higher uncertainty for the first appearance of the Aurignacian and, particularly, the Châtelperronian in this region, which could be older than that estimated with the Bayesian age models.

In the Supramediterranean region, the end of the Mousterian according to the OLE is ~40 ka cal BP (41.93-39.61 at 95%CI), which is consistent with the chronological range obtained from the Bayesian age model (42.76-40.48 ka cal BP). However, it should be noted that, if the resampling procedure is used in the OLE model, the chronological interval at 95.4% CI is larger (41.79-38.19 ka cal BP) than that obtained from the Bayesian age model. This suggests that a higher uncertainty should be acknowledged for the end of the Mousterian in the Submediterranean region.

In the Mesomediterranean region, the end of the Mousterian (41.16-39.87 ka cal BP at 95%CI), and the start (44.10-42.40 ka cal BP) and end (36.25-35.54ka cal BP) of the Aurignacian according to the OLE method are consistent with the chronological ranges obtained from the Bayesian age models (Table 1). Even once the resampling procedure is performed, the start and end of each culture remain quite similar. Likewise, the end of the Mousterian in the Thermomediterranean region (35.43-32.87 ka cal BP) is consistent between the OLE method and the Bayesian age models too.

The more significant difference between the chronology obtained from the Bayesian and the OLE models is observed in the case of the Châtelperronian in the Thermomediterranean area. According to the OLE method, in this region, the Châtelperronian started between 44.14 and 39.71 ka cal BP and ended between 24.94 and 15.73 ka cal BP. These estimations are not meaningful because the Châtelperronian in the Thermomediterranean is represented by only one archaeological site (Cova Foradada) and a low sample size of dates ($n=4$). Therefore, as the Châtelperronian in the Thermomediterranean region is represented by only one archaeological assemblage, the use of Bayesian age modelling is more suitable in this case.

To sum up, the use of the OLE method highlights the uncertainties in the chronological assessment of the Middle to Upper Palaeolithic transition in Iberia, particularly when the number of

radiocarbon determinations is low and when the standard errors of the calibrated dates are large and close to the limit of the calibration curve. Nevertheless, it should be emphasized that outcomes obtained from the OLE method provide greater support to the main results of this study:

- a) The lower limit at 95% CI for the last appearance of the Mousterian in the Eurosiberian region is 43.88 ka cal BP; accordingly, the probability that the Mousterian techno-complex survived the significant drop of the NPP during the GS-12 in this region is lower than 5%. Thus, despite the Neanderthal population decline probably started before the GS-12 (this aspect will be further discussed in the next response to Reviewer 1), both the Bayesian and the OLE models show that the disappearance/LA of the Mousterian culture was coeval to the drop in the NPP during the GS-12.
- b) The Middle to Upper Palaeolithic transition occurred first in the Eurosiberian region, followed by the Mesomediterranean and Thermomediterranean regions (as suggested not only by the Bayesian and the OLE models, but also by previous authors (Higham et al., 2014; Marín-Arroyo et al., 2018; Rios-Garaizar et al., 2022; Zilhão et al., 2017)). Accordingly, there is compelling evidence to propose that the Mousterian lasted longer in the biogeographic regions with more stable NPP.

The use of the OLE method proposed by Reviewer 1 provides a more robust and complete assessment of the first and last appearance of each techno-complex in Iberia. Moreover, to the best of our knowledge, this is the first time this approach is used to assess the chronology of the Middle to Upper Palaeolithic in Iberia. So far, the Middle to Upper Palaeolithic transition in Iberia has been assessed with Bayesian age models. Therefore, we consider that it would be important to keep the results obtained from the Bayesian age models in the main text too, as far as it would facilitate the comparison with the OLE outcomes. Moreover, the use of Bayesian age models is more appropriate than the OLE method for the Châtelperronian in the Thermomediterranean region, as explained above. Thus, the new version of the manuscript includes a brief explanation of the limitations/assumptions of each approach (page 10, lines 445-492) and the results of both methods.

Despite the consensus on the robustness of the OLE method for estimating the FA and LA of Palaeolithic cultures, it should be noted that this procedure shares some assumptions and limitations with the Bayesian models. For instance, OLE assumes a continuation of the phenomena in question after or before the latest or earliest currently known occurrences (this assumption is not valid, for example, in cases of rapid extinction like those derived from catastrophic events or large-scale migrations). Moreover, as in the Bayesian age models, the OLE method does not consider discontinuities in the fossil record, which is one of the concerns raised by Reviewer 1. Therefore, we also analysed the non-modelled dates with Summed Probability Distributions, as proposed by Reviewer 1.

2. Summed Probability Distribution (SPD)

We carried out a Summed Probability Distribution (SPD) analysis, which is commonly used to assess the timing of population/culture changes through time. Despite the large number of scientific researches based on SPDs (Balsera et al., 2015; López de Pablo et al., 2019; Weiberg et al., 2019; Wright et al., 2020), this method is subject to certain limitations and assumptions as well. The main assumption is that population size has a positive correlation with the number of dates, sites, or assemblages. On the other hand, it also assumes that the intensity of research and the preservation of archaeological sites is nearly uniform across the region/s under study. These assumptions are not commonly met, so different filtering steps and statistical procedures have been proposed to overcome these issues and ensure the SPDs as a robust method to estimate population dynamics in prehistory.

Following the recommendations of previous research, the original dataset of radiocarbon dates was filtered. First, dates with large error ranges were removed by eliminating all radiocarbon determinations with a coefficient of variation equal to or larger than 0.05 (Clark et al., 2019). On the other hand, to ensure that each occupational unit is not overrepresented, SPD estimations were based on the chronological distribution of each archaeological assemblage, and dates from the same archaeological layer were merged with the “*R_Combine*” function from OxCal before calibration (López de Pablo et al., 2019). Radiocarbon dates were calibrated with the IntCal20 curve and the SPDs were performed with the *rcarbon* package (Crema & Bevan, 2021) in R. The codes and data of these analyses are also available at <https://github.com/Marco-Vidal/Data-and-codes-associated-with-Vidal-Cordasco-et-al.-Nature-Ecology-Evolution-> The figure below summarises the results obtained (Figure 1).

It can be observed a systematic consistency between the chronological ranges obtained with the Bayesian age models, from the OLE models, and from the summed probability distribution (SPD) of archaeological assemblages to estimate the end/LA of the Mousterian in all regions (Figure 1). This consistency is in agreement with previous research that shows a similarity between the duration of a culture obtained with the “*Boundary*” function in OxCal (not with the “*Date*” function, as explained above) and the duration according to SPDs or Kernel density estimations (Ramsey, 2017).

Reviewer 1 considers that non-modelled dates provide robust and less biased results, so Reviewer 1 suggests including this approach in the main results of the paper. We followed this recommendation and included the SPD of each technocomplex in the Results section of the manuscript. Nevertheless, it is important to note that the OLE method and the SPD shed light on different aspects: the OLE model asks the question: given known artefactual occurrences and past sampling effort, when would the next artefact in the sequence be expected?; on the contrary, SPD reveals the changes in the occurrence frequencies of archaeological assemblages through time. As

both approaches provide valuable and complementary information, we kept both of them in the main results.Access This file is licensed under a Creative Commons Attribution 4.0 International License, which permits use, sharing, adaptation, distribution and reproduction in any medium or format, as long as you give appropriate credit to the original author(s) and the source, provide a link to the Creative Commons license, and indicate if changes were made. In the cases where the authors are anonymous, such as is the case for the reports of anonymous peer reviewers, author attribution should be to 'Anonymous Referee' followed by a clear attribution to the source work. The images or other third party material in this file are included in the article's Creative Commons license, unless indicated otherwise in a credit line to the material. If material is not included in the article's Creative Commons license and your intended use is not permitted by statutory regulation or exceeds the permitted use, you will need to obtain permission directly from the copyright holder. To view a copy of this license, visit <http://creativecommons.org/licenses/by/4.0/>.

Figure 1. Temporal evolution of the mean value (black line) and two times the standard deviation (in shaded grey) of the Net Primary Productivity (NPP) in each biogeographic region between 54.5 and 30 ky BP, with the age probability distribution at 95% CI of the archaeological assemblages and the Summed Probability Distribution (SPD) of these assemblages. The vertical yellow shaded bars show the timing of the end of the Mousterian in each region.

2. The second issue that I would like to highlight relates to the link between the end of Mousterian and drop in productivity (NPP and herbivore carrying capacity). When I look at your figure 4 and the probability distribution for the end of Mousterian in the Eurosiberian region, I can't see a link between the end of Mousterian and the drop of productivity during the GS-12. Given the probability distribution of the Mousterian end, the end likely occurred before 45ka. Although possible based on the confidence limits, it is much more unlikely that the end would have occurred after 45ka. Therefore the link between the Mousterian end and the drop in NPP does not look very strong in the figure 4. The figure 5 might also be a little misleading, because you have extended the Mousterian up to the end of GI-12 (44.28 ka) in the Eurosiberian region, although it is quite unlikely that it lasted so long according to your analyses, as explained above. So my question is, how justifiable is the link between the end of Mousterian and the drop in productivity in the Eurosiberian region? Interestingly, in the nearby Submediterranean region, Mousterian somehow survived the dramatic drop in productivity during GS-12 and disappeared during the next stadial. However, your result might be influenced (biased) by Bayesian age modeling and the related ignorance of archaeological formation processes that I highlighted in the comment 1. I nevertheless find it a really interesting result that Mousterian and maybe Neanderthals themselves seem to disappear earlier in the regions, where fluctuations in productivity are stronger.

R3. Results obtained from the SPD show that, in the Eurosiberian region, the density of Mousterian assemblages declined around 45 ka BP (GI-12) and disappeared during the GS-12 (Figure 1, above). Therefore, despite all the approaches used (SPD, OLE, and Bayesian modelling) show that the end (LA) of the Mousterian techno-complex was during the GS-12, and therefore, coeval with a significant drop of the NPP in the Eurosiberian region, SPD clearly shows that caution is needed when interpreting that drop as the cause of the Neanderthal extinction. Thus, if we assume SPD as a valid proxy of demographic density, the observed Neanderthal population decline started before the GS-12, as Reviewer 1 claims. Yet, this does not invalidate the initial results of the paper. Independently of when Neanderthal populations started to decline, the new chronological analyses support the contention that the Eurosiberian region during the Middle to Upper Palaeolithic

transition witnessed a significant drop in the productivity of the ecosystems, and that likely affected the last Neanderthals in the region.

In the previous version of the manuscript, we stated that “this study does not prove causality between environmental changes and Neanderthal population demise” (previous lines 243-244). In the new version of the manuscript, we emphasise that it cannot be drawn direct causal links between the end of Mousterian and the drop of productivity in the Eurosiberian region, since SPD results suggest that the Neanderthal population decline started before the GS-12 (during the GI-12). Therefore, this drop in the ecosystems productivity provides insight into an ecological factor that probably contributed to their early disappearance in this region, but it cannot be interpreted as a direct and single cause of their demise (page 6, lines 242-247).

To improve the clarity of the results and the comparability between Figures 4 and 5, in the new Figure 5 the duration bar of each techno-complex was modified to incorporate (with dashed bars) the time interval between the start of the decline in the frequency of archaeological assemblages and the final disappearance of each culture according to the SPD.

We thank Reviewer 1 for the insightful comments and for suggesting specific alternative methodologies to assess the chronology of the Middle to Upper Palaeolithic transition, which improved the robustness of the chronological aspects of the study.

Reviewer #2 (Remarks to the Author):

1) This article uses diverse data to evaluate the hypothesis that fluctuations in biomass availability affected the differential disappearance of Neanderthal populations in the Iberian Peninsula. The aspects of the paper that touch on calculating NPP and carrying capacity are outside my direct areas of expertise to the editor will need to rely on the comments of other reviewers when it comes to those aspects.

While I find this paper to be innovative in that it is examining the question of Neanderthal disappearance in Iberia with modified approaches compared to what has been done in the past (e.g. d’Errico and Sanchez Goni 2003 QSR; Banks et al. 2008, which curiously is not cited since they clearly showed that climatic fluctuations were not the main driver behind Neanderthal disappearance), I find that it falls a bit short in laying out it’s chronological foundation, a foundation which is all important for this type of study and for which we have much improved methods today.

R.1 We thank Reviewer 2 for considering innovative the approaches used in this study. Regarding the references suggested by Reviewer 2, d’Errico and Goñi 2003 was already cited in the previous version of the manuscript (previous ref. 16); however, we exceeded the number of references

18recommended by the journal, so we did not include some pertinent papers. In any case, in the new version of the manuscript, we included the reference of Banks et al. 2008 as well (ref. 49 in the new version of the manuscript).

Regarding the concerns with the chronological aspects of the paper, we agree with Reviewer 2: the methods for the chronological transitions of Palaeolithic cultures have much improved in recent years and the previous version of this study did not incorporate these more recent approaches for assessing the first/last appearance of each Palaeolithic technocomplex. Reviewer 1 also pointed out this shortcoming and, in the new version of the manuscript, we analysed the timing of the Middle to Upper Palaeolithic transition by using three different approaches. This was discussed in detail in the responses to Reviewer 1, and we will discuss these methods and results further in the following responses to Reviewer 2.

2) First, I take the current correlations between the end/beginning of an archaeological phase (e.g., end of Mousterian) in a particular region and a particular Greenland Interstadial or Stadial with a bit of caution and this for two reasons. First is the reliance on OxCal. We know that OxCal artificially compresses chronological intervals (Posterior Distribution Functions) based on how it treats dates contained within a defined phase. This is based on its use of the NSBC prior and the fact that it places two hyperparameters at the beginning and end of each phase and the effect of this is that it tends to pull the PDF around the date in a phase that has the smallest standard error (Lanos and Philippe 2018, Communications for Statistical Applications and Methods)—this is especially true for ages that have large standard errors and is visible in almost any study on the late MP or early UP that employs OxCal. The software used by Lanos and Philippe, named ChronoModel, does not use the NSBC prior and only places hyperparameters at the beginning and end of the entire age model, such that it better manages chronological data and their distribution within each phase. Therefore, I would suggest that the authors test their OxCal model against one produced with ChronoModel to test their initial results and evaluate whether they are working with a chronology that is artificially precise. It is freely available at <https://chronomodel.com> and one can also find references to a number of articles that have employed it, as well as to those that describe its mathematics.

R. 2. Reviewer 2 is right when mentioning the limitations of the OxCal software, and we should take with caution the associations between the start/end of archaeological cultures and the specific stadial/interstadial phase because of the uncertainties in the chronological assessments (this is acknowledged in the new version of the manuscript, page 6, lines 267-280). We agree with Reviewer 2 and consider that, to have a robust foundation of the chronological aspects of this study, the chronological assessment of the Middle to Upper Palaeolithic transition should not rely on the use of only one approach/methodology. In the new version of the manuscript, we re-assessed the chronology of the appearance and disappearance of each culture with:

191) Optimal Linear Estimations (OLE), a methodology explicitly developed to assess the first/last appearance of species and recently used with similar purposes in Palaeolithic research (for details, see responses to Reviewer 1),

2) Summed Probability Distribution (SPD), which informs not only about the duration of each culture but also about the changes in the occurrence frequencies of archaeological assemblages through time.

3) The previous Bayesian age models. If we compare the outcomes obtained from OxCal and that obtained with the OLE method (summarised in the table above, Table 1), we see that OxCal may produce a bit younger and particularly shorter chronological intervals for the first appearance of some cultures. However, it should be noted that, even with these uncertainties, OLE models and the SPD approach provide strong consistency to the previous results, as discussed in the responses to Reviewer 1.

Reviewer 2 suggests another alternative approach and asks us to test our OxCal model against one produced by the ChronoModel software. We followed the suggestion of Reviewer 2 and tested our results with the chronology of each culture in the Eurosiberian region by using ChronoModel. The results obtained are summarised below:

Mousterian

49958 – 43440 Date Cal. BP at 95% CI

Châtelperronian

42772-41572 Date Cal. BP at 95% CI

Aurignacian

40645-39243 Date Cal. BP at 95% CI

It can be observed that outcomes obtained from ChronoModel are, overall, in agreement with the SPD, the OLE method, and the previous Bayesian age modelling approach, but the chronological intervals at 95% CI are shorter than those obtained with the OLE method (Table 1). This is not an unexpected result, since the ChronoModel software was developed to assess the chronological range of archaeological assemblages composed by artefacts assumed to be contemporaneous (Llanos & Philippe, 2018); however, in this study the chronology of each culture is composed by different not contemporaneous assemblages. Thus, the dates included as input in ChronoModel (the same happens with OxCal) are not uniformly distributed (as shown with the SPD). On the other hand, ChronoModel highlights the duration of each culture, whereas the OLE model estimates the first/last appearance of each culture. Thus, to estimate the start/end of the Palaeolithic cultures in each region, the OLE method is more suitable and provides more robust results with wider chronological ranges because, unlike the Bayesian statistical approaches, the OLE method does not penalise outlying data (A. Key et al., 2021b). Besides, our main concern with the use of the ChronoModel method is that it did not incorporate the InCal20 curve yet (the most recent available curve in ChronoModel is the IntCal13), so the outcomes obtained from ChronoModel are not fully comparable with the previous results. That said, we feel that this specific concern raised by Reviewer 2 has been addressed with the use of two additional alternative approaches (OLE and SPD), which provide, in the new version of the manuscript, weight to the chronological aspects of the study by revealing the chronological degree of uncertainty.

3. The second issue associated with correlating the Iberian record to climatic chronologies is related to that of radiocarbon calibration and making links to Greenland calendar year paleoclimatic records. One article that details this problem is Giaccio et al. (2017, Sci. Reports) and their examination of how IntCal09 and 13 underestimated real ages, something that is relatively resolved with IntCal20 (Bard et al, 2020 PNAS). This issue, though, is not addressed in the article, so it would be good to see that corrected in order to add weight to the foundational chronological aspects of this study. Finally, it would be good to see a discussion of why the correlations between archaeological intervals or transitions and Greenland climatic chronology are thought to be robust, especially considering the fact that the maximum counting errors for the Greenland record are not anecdotal (ca. 1500-1700 years for the late MP and initial UP; Rasmussen et al. 2014, QSR).

R.3 We completely agree with Reviewer 2: the uncertainties for the associations between the Greenland paleoclimatic records, local or regional paleoclimatic events and archaeological evidences should be acknowledged/discussed in the manuscript. As Reviewer 2 points out, the underestimated ages obtained from the IntCal09 and 13 have been resolved with the InCal20 curve, which was used in this study. However, we still have two main sources of chronological uncertainty: 1) one related to the archaeological intervals, and 2) a second one related to the chronological ranges/counting errors of the stadial/interstadial phases.

21Regarding the uncertainties related to the chronology of the stadial/interstadial phases, this would not alter the main results of this study because we do not pretend to correlate the stadial/interstadial phases with the archaeological intervals, but the NPP fluctuations (reconstructed continuously from 55 to 30 ka BP) with the archaeological intervals. Importantly, there is a match between the fluctuations in the reconstructed NPP and the stadial/interstadial phases as proposed by Rammussen et al. 2014, which suggests a good correspondence between the chronology of the stadial/interstadial phases and the moments of increment/decrease in the ecosystems productivity (Table 1 of the manuscript). Nevertheless, we agree with Reviewer 2, and the error margins of the temporal delimitation of the stadial/interstadial phases are acknowledged in the new version of the manuscript (page 6, lines 269-280)

Regarding the uncertainties of the archaeological intervals (i.e. the appearance/disappearance of each technocomplex), the use of alternative chronological approaches has also revealed the magnitude of these uncertainties. Thus, results obtained from the OLE models show the degree of uncertainty to establish the beginning of some techno-complexes, particularly in the cases with radiocarbon determinations with large standard errors and close to the limit of the calibration curve. This is discussed in the new version of the manuscript (page 6, lines 269-280). However, even if we consider these wide ranges, the main correlations between archaeological intervals and fluctuations in the NPP uphold (Table 1, Figure 1):

-The disappearance of the Mousterian, and the Middle to Upper Palaeolithic transition, in the Eurosiberian region was coeval with a significant drop in the ecosystems productivity, independently of the specific stadial/interstadial phase (this is supported by the OLE, the SPD and both Bayesian approaches -OxCal and ChronoModel-).

-The Mousterian lasted longer in the biogeographic regions with more stable productivity (Mesomediterranean and Thermomediterranean regions) and higher carrying capacity of medium and medium-large herbivores during the stadial phases (also supported by the OLE, the SPD and the Bayesian age models).

4. I do not have an issue with the NPP and biomass aspects of the paper, but if one is to have any confidence in the conclusions that are drawn from the results, we must be entirely convinced that our conclusions are drawn from correct associations between the archaeological record and the corresponding paleoclimatic context, associations that I think merit being more solidly demonstrated.

R.4. We agree on the relevance of the chronological aspects of this study and it should be as robust as possible. We consider that the new version of the manuscript improved by increasing the number of chronological approaches from one (in the previous version) to three. These methodologies incorporated in the new version of the manuscript reveal the degree of uncertainty

to estimate the beginning of some techno-complexes and provide solid support to the main results of the paper.

We wish to thank Reviewer 2 for the comments, which enriched the paper by paying more attention to the uncertainties of the chronological aspects of the study.

Reviewer #3 (Remarks to the Author):

1) This paper assesses the effects of ecosystem productivity (Net Primary Productivity and Total Herbivore Biomass) on extinction of the Neanderthals in four biogeographic regions of Iberia. It is a complex and highly robust paper that combines Bayesian age modelling of the major industries, palaeoclimatic reconstructions, and resultant estimates of NPP and herbivore biomass. Importantly, the paper does not suggest that changes in NPP and herbivore biomass would have effected the two species differently - it simply combines age estimates for start and end dates of the various industries to examine alignments between environmental changes and the decline of Neanderthal (and AMH) populations. The fact that results for the four regions are quite different is of interest, and demonstrates an objective approach (i.e. the authors genuinely set out to test their hypothesis rather than to prove it, which is of course the way it should be).

R.1 We thank Reviewer 3 for the positive feedback.

2) I have no major problems with the paper, but below are a series of questions / suggestions for clarification:

Line 86 (etc.) You refer to 'moments' - maybe 'periods' or 'phases' would be better.

R.2 "Moments" was replaced by "periods" or "phases" throughout the manuscript (lines 60, 84, 96, 100, 135, 639, 648)

3) Line 159ff. The Eurosiberian region is the only one in which climate seems to play a key role - it would be useful to explicitly mark the decrease in resource availability on Figure 3.

R.3 Following the suggestion of Reviewer 3, to explicitly mark this association in Figure 3 now we mark with vertical shaded bars the disappearance moment of the Mousterian in each region.

4) Line 244-281. The caveats and limitations are made very clear here, which adds to the overall robusticity of the paper.

R.4 We thank Reviewer 3 for the positive assessment of the caveats discussed in this study. In the new version of the manuscript, the discussion of these limitations was extended by acknowledging the uncertainties with the chronological aspects, as Reviewers 1 and 2 suggested (page 6, lines 266-285 and page 10, lines 455-493).

5) I couldn't find anything about how radiocarbon dates are treated when they come from the same site (or same stratigraphic layer). Presumably multiple dates on the same stratigraphic layer are averaged in some way before being used in the Bayesian models, otherwise there is a risk that sites that have had extensive data programmes are over-represented (in other words, the results end up being skewed by research effort). Could you discuss this briefly (preferably in the main 'Methods' text).

R.5 The chronological aspect of the study experienced a substantial improvement in the new version of the manuscript. Thus, following the suggestions of Reviewers 1 and 2, we incorporated two more alternative methodologies to assess the duration of each archaeological culture: Optimal Linear Estimations (OLE), and Summed Probability Distributions (SPD). These approaches have their specific assumptions, limitations, filtering processes, and protocols, including the averaging of radiocarbon dates obtained from the same layer if necessary. In the SPD, multiple dates obtained from the same stratigraphic layer are commonly averaged, and this was performed in this study by using the “*Combine*” function in OxCal before using these input data in the SPD calculations. In this way, there are not assemblages overrepresented nor biased by research effort. On the contrary, in the Bayesian age models this “*Combine*” function is not used to merge the dates obtained from the same level, but when different dates were obtained from the same bone. The reason for this differential treatment is because SPD assess the distribution of occupation frequencies/intensity of archaeological assemblages through time, so this averaging procedure avoids the research intensity bias suggested by Reviewer 3; on the contrary, the Bayesian age models assess the duration or boundaries of phases (not the intensity/frequency of occupation), so this averaging process of all radiocarbon dates is not commonly used. Following the suggestion of Reviewer 3, this is detailed in the new version of the Methods section (page 9, lines 403-493).

6) Equations 1 & 2: What are S_1 , S_2 , p , u , and v ? Presumably S_1 and S_2 are time series 1 and 2, so is p the length of the time series, with u and v values from each time series? I'm guessing that this is what is happening, but it would be better not to have to guess. Likewise, what is a 'raw-data distance'? Is this a Euclidean distance? Is it total distance between the two whole series, or for a series of pairwise points?

24R6. S_1 and S_2 are the two series of data. Thus, dCORT assesses the similarity between the datasets S_1 and S_2 . On the other hand, Reviewer 3 understood it correctly: u represents the values of S_1 , v the values of S_2 , and p is the number/length of values of each time series. Therefore: $S_1 = (u_1, u_2, \dots, u_p)$ and $S_2 = (v_1, v_2, \dots, v_p)$. Regarding the 'raw-data distance', this is the difference between each series of pairwise points. All these aspects have been clarified in the new version of the manuscript (page 9, lines 373-386)

7) Equation 3: What value of the parameter k is actually used? How and why was this value chosen?

R7. The value of the parameter k used is 2. The K value defines the contribution of the behaviour and the contribution of the proximity between values when comparing the datasets. If $k=0$, the % contribution of the proximity in behaviour is 0 and the contribution of the proximity between values is 100% (Montero & Vilar, 2015). In this study, we used the default value of 2 used in the R package *TsClust*. This means that the behaviour contribution is 76% and the contribution of the proximity between values is 24% (see (Montero & Vilar, 2015) for details). It should be noted that if k value was 1 (in this case, the % contribution of the behaviour proximity would be 46% and that of the values 54% (Montero & Vilar, 2015)), we would obtain the same three main groups, as it can be observed in the following Figure 2. In the new version of the manuscript, we specify the value of k (page 9, lines 385-386).

K=2

K=1

Figure 2. Dendrogram showing the clustering association of the NPP estimated in the

25

surrounding area of each archaeo-paleontological site based on the dCORT dissimilarity index with a k value of 2 (left) and 1 (right).

8) Equations 5 and 6 have very different forms, yet they are delta corrections of temperature and precipitation respectively. Why do they have such different forms?

R8. Both equations (5 and 6) perform a delta correction, but they have different forms because the temperature may have negative values, but precipitations cannot have negative values. Thus, the formulation of equation 6 avoids the risk of getting negative values after the bias-correction procedure.

9) Finally, a couple of points that require clarification in terms of the faunal evidenc. What if it is not overall THB that is important, but the abundances of particular species (and what if this differs between Neanderthals and AMH)? Since you have the data, why not consider this in relation to faunal assemblages from Neanderthal and AMH sites? Are there any differences? Are any species particularly over-represented?

R.9 This is a quite interesting point, but it should be noted that the results presented in the manuscript were not restricted to the overall THB. Comparisons between specific species abundances between regions are difficult not only because of the relatively large number of mammal species but also because the species composition differed between biogeographic regions and moments of the MIS 3. Thus, not all species are present in all these biogeographic regions during the whole MIS 3. For that reason, we analysed the species abundances by grouping species per size category: small, medium, medium-large, and large (as presented in Fig 5 and explained in Methods). In this connection, we showed that the abundance of medium and medium-large mammals probably was particularly important for the Neanderthal population persistence in the Thermomediterranean region, because there was a higher carrying capacity of these herbivores in southern latitudes in comparison with the Eurosiberian region during the cold stadial phases of the MIS 3. The medium-large species would include *Ovibos moschatus*, *Equus ferus*, *Equus hydruntinus*, *Cervus elpahus*, and *Sus scrofa*. On the other hand, medium-sized herbivores would include: *Rangifer tarandus*, *Capreolus capreolus*, *Capra pyrenaica*, *Capra ibex*, *Dama dama*, *Rupicapra pyrenaica*, *Rupicapra rupicapra*. The importance of each one of these species to the diet of AMH or Neanderthals probably varied depending on several factors, such as the location of the site. Thus, the abundance or relevance of these species separately would be influenced by many different variables, such as the specific orography of the site or the season of the occupation. This approach would be interesting, but it would require focussing on specific smaller areas, which is

beyond the scope of the current study. Due to the limit of words for the Main text according to the journal guidelines, we briefly state in the Discussion section that further research should focus on specific species and smaller areas (page 6, lines 283-285).

10. Are you only using archaeological faunal assemblages as indicators of the available faunal species at the time? This approach is useful in that it directly reflects subsistence practices, but of course it is not a reflection of overall faunal abundances in the region (because there is likely to be some selectivity in hunting strategies, which biases species representation in archaeological assemblages). It would be good to see this issue discussed (even briefly) in the main Methods text.

R. 10. No, we are using both archaeological and paleontological faunal assemblages whenever they have chronometric determinations. This is stated on page 7, lines 316-317, and page 8, lines 327-330.

We thank Reviewer 3 for the thoroughly analysis of the equations used in this study and the feedback provided, which significantly contributed to the clarity of the manuscript.

REFERENCES

- Balsera, V., Díaz-del-Río, P., Gilman, A., Uriarte, A., & Vicent, J. M. (2015). Approaching the demography of late prehistoric Iberia through summed calibrated date probability distributions (7000-2000 cal BC). *Quaternary International*, 386, 208–211. <https://doi.org/10.1016/j.quaint.2015.06.022>
- Bebber, M. R., & Key, A. J. M. (2022). Optimal Linear Estimation (OLE) Modeling Supports Early Holocene (9000–8000 RCYBP) Copper Tool Production in North America. *American Antiquity*, 87(2), 267–283. <https://doi.org/10.1017/AAQ.2021.121>
- Clark, G. A., Michael Barton, C., & Straus, L. G. (2019). Landscapes, climate change & forager mobility in the Upper Paleolithic of northern Spain. *Quaternary International*, 515, 176–187. <https://doi.org/10.1016/J.QUAINT.2018.04.037>
- Crema, E. R., & Bevan, A. (2021). Inference from large sets of radiocarbon dates: software and methods. *Radiocarbon*, 63(1), 23–39. <https://doi.org/10.1017/RDC.2020.95>
- Higham, T., Douka, K., Wood, R., Ramsey, C. B., Brock, F., Basell, L., Camps, M., Arrizabalaga, A., Baena, J., Barroso-Ruiz, C., Bergman, C., Boitard, C., Boscato, P., Caparrós, M., Conard, N. J., Draily, C., Froment, A.,

- Galván, B., Gambassini, P., ... Jacobi, R. (2014). The timing and spatiotemporal patterning of Neanderthal disappearance. *Nature* 2014 512:7514, 512(7514), 306–309. <https://doi.org/10.1038/nature13621>
- Key, A. J. M., Jarić, I., & Roberts, D. L. (2021). Modelling the end of the Acheulean at global and continental levels suggests widespread persistence into the Middle Palaeolithic. *Humanities and Social Sciences Communications* 2021 8:1, 8(1), 1–12. <https://doi.org/10.1057/s41599-021-00735-8>
- Key, A. J. M., Roberts, D. L., & Jarić, I. (2021). Statistical inference of earlier origins for the first flaked stone technologies. *Journal of Human Evolution*, 154, 102976. <https://doi.org/10.1016/J.JHEVOL.2021.102976>
- Key, A., Roberts, D., & Jarić, I. (2021a). Reconstructing the full temporal range of archaeological phenomena from sparse data. *Journal of Archaeological Science*, 135, 105479. <https://doi.org/10.1016/J.JAS.2021.105479>
- Key, A., Roberts, D., & Jarić, I. (2021b). Reconstructing the full temporal range of archaeological phenomena from sparse data. *Journal of Archaeological Science*, 135, 105479. <https://doi.org/10.1016/J.JAS.2021.105479>
- Llanos, P., & Philippe, A. (2018). Event date model: A robust Bayesian tool for chronology building. *Communications for Statistical Applications and Methods*, 25(2), 131–157.
- López de Pablo, J., Gutiérrez-Roig, M., Gómez-Puche, M., McLaughlin, R., Silva, F., & Lozano, S. (2019). Palaeodemographic modelling supports a population bottleneck during the Pleistocene-Holocene transition in Iberia. *Nature Communications*, 10(1), 1–13. <https://doi.org/10.1038/s41467-019-09833-3>
- Marín-Arroyo, A. B., Rios-Garaizar, J., Straus, L. G., Jones, J. R., de la Rasilla, M., González Morales, M. R., Richards, M., Altuna, J., Mariezkurrena, K., & Ocio, D. (2018). Chronological reassessment of the Middle to Upper Paleolithic transition and Early Upper Paleolithic cultures in Cantabrian Spain. *PLoS ONE*, 13(4), e0194708. <https://doi.org/10.1371/journal.pone.0194708>
- Montero, P., & Vilar, J. A. (2015). TSclust: An R Package for Time Series Clustering. *Journal of Statistical Software*, 62(1), 1–43. <https://doi.org/10.18637/JSS.V062.I01>
- Ramsey, C. B. (2017). Methods for Summarizing Radiocarbon Datasets. *Radiocarbon*, 59(6), 1809–1833. <https://doi.org/10.1017/RDC.2017.108>
- Rios-GaraizarID, J., Iriarte, E., Arnold, L. J., Sá nchez-Romero, L., Marín-Arroyo, A. B., San Emeterio, A., Gómez-Olivencia, A., Pérez-Garrido, C., DemuroID, M., Campaña, I., Bourguignon, L., Benito-Calvo, A., Iriarte, M. J., Aranburu, A., Arranz-Otaegi, A., Garate, D., Silva-GagoID, M., Lahaye, C., Ortega, I., & Museoa, A. (2022). The intrusive nature of the Châtelperronian in the Iberian Peninsula. *PLOS ONE*, 17(3), e0265219. <https://doi.org/10.1371/JOURNAL.PONE.0265219>

Weiberg, E., Bevan, A., Kouli, K., Katsianis, M., Woodbridge, J., Bonnier, A., Engel, M., Finné, M., Fyfe, R., Maniatis, Y., Palmisano, A., Panajiotidis, S., Roberts, C. N., & Shennan, S. (2019). Long-term trends of land use and demography in Greece: A comparative study. *The Holocene*, 29(5), 742–760. <https://doi.org/10.1177/0959683619826641>

Wright, D. K., Kim, J., Park, J., Yang, J., & Kim, J. (2020). Spatial modeling of archaeological site locations based on summed probability distributions and hot-spot analyses: A case study from the Three Kingdoms Period, Korea. *Journal of Archaeological Science*, 113, 105036. <https://doi.org/10.1016/J.JAS.2019.105036>

Zilhão, J., Anesin, D., Aubry, T., Badal, E., Cabanes, D., Kehl, M., Klasen, N., Lucena, A., Martín-Lerma, I., Martínez, S., Matias, H., Susini, D., Steier, P., Wild, E. M., Angelucci, D. E., Villaverde, V., & Zapata, J. (2017). Precise dating of the Middle-to-Upper Paleolithic transition in Murcia (Spain) supports late Neandertal persistence in Iberia. *Heliyon*, 3(11). <https://doi.org/10.1016/J.HELIYON.2017.E00435>

Decision Letter, first revision:

14th June 2022

Dear Dr Marín-Arroyo,

Your manuscript entitled "Ecosystems productivity affected the spatiotemporal disappearance of Neanderthals in Iberia" has now been seen by the three original reviewers, whose comments are attached. The reviewers have raised a number of concerns which will need to be addressed before we can offer publication in *Nature Ecology & Evolution*. We will therefore need to see your responses to the criticisms raised and to some editorial concerns, along with a revised manuscript, before we can reach a final decision regarding publication.

It looks like we are very close to a publishable version, but we will need to see you address the remaining comments from reviewer 2 regarding the need to justify use of summed probability distributions (and note that reviewer 1 also refers to this in passing, though they see less of an issue with the use)--this should be addressable with discussion and caveats rather than changes to the methods.

29We therefore invite you to revise your manuscript taking into account all reviewer and editor comments. Please highlight all changes in the manuscript text file [OPTIONAL: in Microsoft Word format].

* If you have not done so already please begin to revise your manuscript so that it conforms to our Article format instructions at <http://www.nature.com/natecolevol/info/final-submission>. Refer also to any guidelines provided in this letter.

[REDACTED]

Nature Ecology & Evolution is committed to improving transparency in authorship. As part of our efforts in this direction, we are now requesting that all authors identified as 'corresponding author' on published papers create and link their Open Researcher and Contributor Identifier (ORCID) with their account on the Manuscript Tracking System (MTS), prior to acceptance. ORCID helps the scientific community achieve unambiguous attribution of all scholarly contributions. You can create and link your ORCID from the home page of the MTS by clicking on 'Modify my Springer Nature account'. For more information please visit www.springernature.com/orcid.

30We look forward to seeing the revised manuscript and thank you for the opportunity to review your work.

[REDACTED]

Reviewer expertise:

as before

Reviewers' comments:

Reviewer #1 (Remarks to the Author):

Authors have responded well to reviewers comments and have been able to further improve their paper by including OLE and SPD methods. This has strengthened the main arguments including the one about the link between drops in productivity and human population decrease/extinction. Although some degree of uncertainty remains regarding the link, I would not consider that as a significant issue (there will always be in paleoresearch). Thus, authors have carefully revised all the issues that I raised in the first round of reviews of this interesting and important contribution.

Reviewer #2 (Remarks to the Author):

It is good to see that the authors have taken into account the comments from the various reviewers, especially concerning the chronological aspects of the manuscript and, as it concerns chronology, they present results derived from a number of methods in an effort to validate and render more robust their conclusions.

That being said, and even though ChronoModel results were not retained, I get the impression that the authors are unclear as to how ChronoModel and the concept of "Event" work, which would be useful to correct if they are to use this approach in the future. In ChronoModel dated samples are not assumed to be contemporaneous. Each dated sample is a different event (unless two ages come from the same sample or are shown to be unequivocally archaeologically contemporaneous) and events are not contemporaneous—Events can be grouped into phases when they are associated with the same archaeological culture but they are not contemporaneous. This is how it differs mathematically from OxCal. These are moot points however since the authors did not integrate the IntCal20 curve data into CM and thus did not retain it for the article.

For the chronological methods that were retained, I have reservations about their use of Summed Probability Distributions as it stands now since the known limitations of such an approach are not discussed. That is not to say that some modification of such an approach could not be used, but I do

31not see that the authors present their reasons for using the method, nor what they have done to make sure that its use is appropriate. This could be easily corrected by looking at recent literature on the topic (which is not cited in the manuscript) and modifying this aspect of the study so that it is up-to-date with the current state of the art (e.g., Crema and Bevan 2021 Radiocarbon; Price et al. 2021 J. Arch. Sci).

If this issue is dealt with then I do not have any objection to the article being published, although I am not entirely convinced by their conclusions concerning the Ebro frontier. However, if their findings are based in the end on solid chronological results, then it is up to the scientific community to weigh in on their conclusions, rather than just reviewers.

Reviewer #3 (Remarks to the Author):

The authors have done an admirably thorough job in responding to my questions, and have also improved the manuscript in relation to comments made by the other reviewers. Some issues of interpretation remain, but these should not prevent publication, as they relate to methodological debates that are ongoing and have no obvious optimal resolution. I am therefore convinced that the paper now merits publication.

*****END*****

Author Rebuttal, first revision:

Reviewer #1 (Remarks to the Author):

Authors have responded well to reviewers comments and have been able to further improve their paper by including OLE and SPD methods. This has strengthened the main arguments including the one about the link between drops in productivity and human population decrease/extinction. Although some degree of uncertainty remains regarding the link, I would not consider that as a significant issue (there will always be in paleoresearch). Thus, authors have carefully revised all the issues that I raised in the first round of reviews of this interesting and important contribution.

We appreciate very much that Reviewer 1 considers interesting and important the contribution made in this work. Moreover, we are happy of having resolved all the concerns raised by Reviewer 1 in the previous round of reviews.

32Reviewer #2 (Remarks to the Author):

It is good to see that the authors have taken into account the comments from the various reviewers, especially concerning the chronological aspects of the manuscript and, as it concerns chronology, they present results derived from a number of methods in an effort to validate and render more robust their conclusions.

That being said, and even though ChronoModel results were not retained, I get the impression that the authors are unclear as to how ChronoModel and the concept of “Event” work, which would be useful to correct if they are to use this approach in the future. In ChronoModel dated samples are not assumed to be contemporaneous. Each dated sample is a different event (unless two ages come from the same sample or are shown to be unequivocally archaeologically contemporaneous) and events are not contemporaneous—Events can be grouped into phases when they are associated with the same archaeological culture but they are not contemporaneous. This is how it differs mathematically from OxCal. These are moot points however since the authors did not integrate the IntCal20 curve data into CM and thus did not retain it for the article.

For the chronological methods that were retained, I have reservations about their use of Summed Probability Distributions as it stands now since the known limitations of such an approach are not discussed. That is not to say that some modification of such an approach could not be used, but I do not see that the authors present their reasons for using the method, nor what they have done to make sure that its use is appropriate. This could be easily corrected by looking at recent literature on the topic (which is not cited in the manuscript) and modifying this aspect of the study so that it is up-to-date with the current state of the art (e.g., Crema and Bevan 2021 Radiocarbon; Price et al. 2021 J. Arch. Sci).

If this issue is dealt with then I do not have any objection to the article being published, although I am not entirely convinced by their conclusions concerning the Ebro frontier. However, if their findings are based in the end on solid chronological results, then it is up to the scientific community to weigh in on their conclusions, rather than just reviewers.

Reviewer 2 is right regarding our previous comments about the limitations of ChronoModel. Actually, after a carefully re-reading of the paper presenting the ChronoModel software, we think that if it would incorporate the IntCal20 curve it would be an ideal methodology to assess the

regional chronology of Palaeolithic cultures. We look forward to the software developers clarifying how to incorporate the IntCal20 curve to use this modelling approach in future studies.

We followed the suggestion of Reviewer 2 and, in the new version of the manuscript, we highlight the limitations of Summed Probability Distributions (SPD) and specify the reason of using such an approach (namely, assessing the occurrence frequency of archaeological assemblages over time in each region) and what we have done to make sure that its use is appropriate. Thus, we incorporated the following statements in the Discussion sections:

“On the other hand, despite the popularity of SPD, it should be noted that frequency of radiocarbon dates is not a direct indicator of population sizes, and therefore, it should not be used as a pristine demographic proxy⁵³ (see the Method section for details).” (page 6, lines 279-282)

And in greater details, in the Method section:

“SPD is commonly used as a demographic proxy in Palaeolithic research. However, this method is subjected to certain assumptions^{53,63}. SPD analyses assume that population size has a positive correlation with the number of dates, sites, or assemblages, and that the intensity of research and preservation of archaeological sites is nearly uniform across the region of study. These assumptions are not commonly met, so some filtering steps are necessary to ensure that SPD is a robust method to infer population dynamics. Following the recommendations of previous research⁶³, several filtering processes were used. First, dates with large error ranges were removed by eliminating all chronometric determinations with a coefficient of variation equal to or larger than 0.05⁶⁴. Second, to ensure that each occupational unit is not overrepresented, SPD estimations were based on the chronological distribution of each archaeological assemblage, and dates from the same archaeological level were merged with the “*R_Combine*” function from OxCal before calibration⁶⁵. Third, radiocarbon determinations obtained from shell remains were excluded because of the uncertainties related to the marine reservoir offsets, and the remaining dates were calibrated with the IntCal20 calibration curve⁵⁵. These filtering steps overcome some of the limitations of this method, but caveats are still necessary⁶³. Therefore, outcomes obtained from the SPD have not been used to inform demographics in this study, but as an additional method to assess the duration and, particularly, the frequency of occupation of each culture.” (page 11, lines 492-509)

Finally, we incorporated the recent literature suggested by Reviewer 2. We hope these modifications resolve the concerns of Reviewer 2 and thank Reviewer 2 for the comments. There is a long-standing debate about the Ebro frontier hypothesis, but we consider that it is important to show that the conclusions of this study are not contingent on the modelling method used to reconstruct the chronology of each culture. Thus, to provide additional support to the links between the NPP and the Middle to Upper Palaeolithic transition in each region, we incorporated one additional Figure in the Supplementary Materials with the NPP and the raw calibrated dates (when

no modelling approach is used) in each region. Accordingly, we also incorporated a brief sentence in the Discussion section:

“In this connection, the comparison of the raw calibrated dates (when no modelling method is used) with the reconstructed NPP in each region suggests that the conclusions of this study do not depend on modelling approach used to estimate the chronology of the Middle to Upper Palaeolithic transition (Supplementary Figure 4).” (page 6, lines 284-287)”

Reviewer #3 (Remarks to the Author):

The authors have done an admirably thorough job in responding to my questions, and have also improved the manuscript in relation to comments made by the other reviewers. Some issues of interpretation remain, but these should not prevent publication, as they relate to methodological debates that are ongoing and have no obvious optimal resolution. I am therefore convinced that the paper now merits publication.

We thank Reviewer 3 for acknowledging the work involved in this research and for considering that the paper merits publication.

Decision Letter, second revision:

4th July 2022

Dear Dr. Marín-Arroyo,

Thank you for submitting your revised manuscript "Ecosystems productivity affected the spatiotemporal disappearance of Neanderthals in Iberia" (NATECOLEVOL-220215841B). It has now been seen again by the original reviewers and their comments are below. The reviewers find that the paper has improved in revision, and therefore we'll be happy in principle to publish it in Nature Ecology & Evolution, pending minor revisions to comply with our editorial and formatting guidelines.

35We are now performing detailed checks on your paper and will send you a checklist detailing our editorial and formatting requirements in about a week. Please do not upload the final materials and make any revisions until you receive this additional information from us.

[REDACTED]

Reviewer #2 (Remarks to the Author):

The authors have adequately addressed the concerns I raised concerning the initially revised submission, and I appreciate their effort to do so. The presently revised manuscript merits publication.

William E. Banks

Our ref: NATECOLEVOL-220215841B

8th July 2022

Dear Dr. Marín-Arroyo,

Thank you for your patience as we've prepared the guidelines for final submission of your Nature Ecology & Evolution manuscript, "Ecosystems productivity affected the spatiotemporal disappearance of Neanderthals in Iberia" (NATECOLEVOL-220215841B). Please carefully follow the step-by-step instructions provided in the attached file, and add a response in each row of the table to indicate the changes that you have made. Please also check and comment on any additional marked-up edits we have proposed within the text. Ensuring that each point is addressed will help to ensure that your revised manuscript can be swiftly handed over to our production team.

****We would like to start working on your revised paper, with all of the requested files and forms, as soon as possible (preferably within two weeks). Please get in contact with us immediately if you anticipate it taking more than two weeks to submit these revised files.****

36If you have not done so already, please alert us to any related manuscripts from your group that are under consideration or in press at other journals, or are being written up for submission to other journals (see: <https://www.nature.com/nature-research/editorial-policies/plagiarism#policy-on-duplicate-publication> for details).

In recognition of the time and expertise our reviewers provide to Nature Ecology & Evolution's editorial process, we would like to formally acknowledge their contribution to the external peer review of your manuscript entitled "Ecosystems productivity affected the spatiotemporal disappearance of Neanderthals in Iberia". For those reviewers who give their assent, we will be publishing their names alongside the published article.

Nature Ecology & Evolution offers a Transparent Peer Review option for new original research manuscripts submitted after December 1st, 2019. As part of this initiative, we encourage our authors to support increased transparency into the peer review process by agreeing to have the reviewer comments, author rebuttal letters, and editorial decision letters published as a Supplementary item. When you submit your final files please clearly state in your cover letter whether or not you would like to participate in this initiative. Please note that failure to state your preference will result in delays in accepting your manuscript for publication.

Cover suggestions

As you prepare your final files we encourage you to consider whether you have any images or illustrations that may be appropriate for use on the cover of Nature Ecology & Evolution.

Nature Ecology & Evolution has now transitioned to a unified Rights Collection system which will allow our Author Services team to quickly and easily collect the rights and permissions required to publish your work. Approximately 10 days after your paper is formally accepted, you will receive an email in providing you with a link to complete the grant of rights. If your paper is eligible for Open Access, our Author Services team will also be in touch regarding any additional information that may be required to arrange payment for your article.

Please note that *Nature Ecology & Evolution* is a Transformative Journal (TJ). Authors may publish their research with us through the traditional subscription access route or make their paper immediately open access through payment of an article-processing charge (APC). Authors will not be required to make a final decision about access to their article until it has been accepted. [Find out more about Transformative Journals](https://www.springernature.com/gp/open-research/transformative-journals)

Authors may need to take specific actions to achieve [compliance with funder and institutional open access mandates](https://www.springernature.com/gp/open-research/funding/policy-compliance-faqs). If your research is supported by a funder that requires immediate open access (e.g. according to [Plan S principles](https://www.springernature.com/gp/open-research/plan-s-compliance)) then you should select the gold OA route, and we will direct you to the compliant route where possible. For authors selecting the subscription publication route, the journal's standard licensing terms will need to be accepted, including [self-archiving and license to publish](https://www.nature.com/nature-portfolio/editorial-policies/self-archiving-and-license-to-publish). Those licensing terms will supersede any other terms that the author or any third party may assert apply to any version of the manuscript.

[REDACTED]

[REDACTED]

Reviewer #2:

Remarks to the Author:

The authors have adequately addressed the concerns I raised concerning the initially revised submission, and I appreciate their effort to do so. The presently revised manuscript merits publication.

William E. Banks

Final Decision Letter:

28th July 2022

Dear Dr Marín-Arroyo,

We are pleased to inform you that your Article entitled "Ecosystem productivity affected the spatiotemporal disappearance of Neanderthals in Iberia", has now been accepted for publication in Nature Ecology & Evolution.

Over the next few weeks, your paper will be copyedited to ensure that it conforms to Nature Ecology and Evolution style. Once your paper is typeset, you will receive an email with a link to choose the appropriate publishing options for your paper and our Author Services team will be in touch regarding any additional information that may be required

You will not receive your proofs until the publishing agreement has been received through our system

Due to the importance of these deadlines, we ask you please us know now whether you will be difficult to contact over the next month. If this is the case, we ask you provide us with the contact information (email, phone and fax) of someone who will be able to check the proofs on your behalf, and who will be available to address any last-minute problems . Once your paper has been scheduled for online publication, the Nature press office will be in touch to confirm the details.

Acceptance of your manuscript is conditional on all authors' agreement with our publication policies (see www.nature.com/authors/policies/index.html). In particular your manuscript must not be published elsewhere and there must be no announcement of the work to any media outlet until the publication date (the day on which it is uploaded onto our web site).

Please note that *Nature Ecology & Evolution* is a Transformative Journal (TJ). Authors may publish their research with us through the traditional subscription access route or make their paper immediately open access through payment of an article-processing charge (APC). Authors will not be required to make a final decision about access to their article until it has been accepted. [Find out more about Transformative Journals](https://www.springernature.com/gp/open-research/transformative-journals)

Authors may need to take specific actions to achieve [compliance](https://www.springernature.com/gp/open-research/funding/policy-compliance-faqs) with funder and institutional open access mandates. If your research is supported by a funder that requires immediate open access (e.g. according to [Plan S principles](https://www.springernature.com/gp/open-research/plan-s-compliance)) then you should select the gold OA route, and we will direct you to the compliant route where possible. For authors selecting the subscription publication route, the journal's standard licensing terms will need to be accepted, including <https://www.nature.com/nature-portfolio/editorial->

39policies/self-archiving-and-license-to-publish. Those licensing terms will supersede any other terms that the author or any third party may assert apply to any version of the manuscript.

We welcome the submission of potential cover material (including a short caption of around 40 words) related to your manuscript; suggestions should be sent to Nature Ecology & Evolution as electronic files (the image should be 300 dpi at 210 x 297 mm in either TIFF or JPEG format). Please note that such pictures should be selected more for their aesthetic appeal than for their scientific content, and that colour images work better than black and white or grayscale images. Please do not try to design a cover with the Nature Ecology & Evolution logo etc., and please do not submit composites of images related to your work. I am sure you will understand that we cannot make any promise as to whether any of your suggestions might be selected for the cover of the journal.

You can generate the link yourself when you receive your article DOI by entering it here: <http://authors.springernature.com/share>.

[REDACTED]

P.S. Click on the following link if you would like to recommend Nature Ecology & Evolution to your librarian <http://www.nature.com/subscriptions/recommend.html#forms>

** Visit the Springer Nature Editorial and Publishing website at http://editorial-jobs.springernature.com?utm_source=ejP_NEcoE_email&utm_medium=ejP_NEcoE_email&utm_campaign=

40www.springernature.com/editorial-and-publishing-jobs for more information about our career opportunities. If you have any questions please click [here](mailto:editorial.publishing.jobs@springernature.com).**